# Stochastic pulsing of gene expression enables the generation of spatial patterns in *Bacillus subtilis* biofilms

Eugene Nadezhdin[1,4], Niall Murphy[1,2,4], Neil Dalchau [2], Andrew Phillips [2] & James C.W. Locke[1,2,3 ✉]

Stochastic pulsing of gene expression can generate phenotypic diversity in a genetically identical population of cells, but it is unclear whether it has a role in the development of multicellular systems. Here, we show how stochastic pulsing of gene expression enables spatial patterns to form in a model multicellular system, *Bacillus subtilis* bacterial biofilms. We use quantitative microscopy and time-lapse imaging to observe pulses in the activity of the general stress response sigma factor $\sigma^B$ in individual cells during biofilm development. Both $\sigma^B$ and sporulation activity increase in a gradient, peaking at the top of the biofilm, even though $\sigma^B$ represses sporulation. As predicted by a simple mathematical model, increasing $\sigma^B$ expression shifts the peak of sporulation to the middle of the biofilm. Our results demonstrate how stochastic pulsing of gene expression can play a key role in pattern formation during biofilm development.

[1] Sainsbury Laboratory, University of Cambridge, Cambridge CB2 1LR, UK. [2] Microsoft Research, 21 Station Rd, Cambridge CB1 2FB, UK. [3] Department of Biochemistry, University of Cambridge, Cambridge CB2 1QW, UK. [4]These authors contributed equally: Eugene Nadezhdin, Niall Murphy. ✉email: james.locke@slcu.cam.ac.uk

Spatial patterns of gene expression play a key role in development[1,2]. Although many patterns are formed in response to spatial gradients of initiating signals[3,4], noise in the form of transient fluctuations in gene expression can also initiate patterns by generating differences in gene expression within a population of identical cells[5–9]. Noise in gene expression has been observed in major regulators at the level of individual bacterial cells[10–12]. This noise is hypothesised to play a functional role, generating phenotypic diversity that can allow the population to bet-hedge against variable environments[13] by promoting a subset of cells to enter a slow growing persistent state that can survive antibiotic treatment[14,15].

Recent work in multiple systems has revealed that phenotypic diversity in a population of genetically identical cells can be achieved through the pulsed activity of transcriptional regulators. Stochastic pulsing of key regulators has been observed in multiple systems, from microbes to mammalian cell culture[16]. However, the role of stochastic pulses of gene expression in the development of multicellular systems remains poorly understood. Here we use *Bacillus subtilis* bacterial biofilms as a model multicellular system to investigate the role of stochastic pulses of gene expression during pattern formation.

Bacterial biofilms[17–19] are complex multicellular systems that exhibit a variety of spatial structures including vasculature[20], protective layers[21,22], and functional specialisation[23]. Snapshots of *B. subtilis* biofilms containing fluorescent reporters for key pathways have revealed the formation of spatial patterns within the biofilm, with heterogeneous cell states for motility, sporulation, surfactin production, and matrix formation[24]. These cell states[25] are often heterogeneously distributed in a "salt and pepper" fashion[26], yet are spatially localised to specific regions of the biofilm[24], suggesting a role for both spatial signalling and noise in gene expression during pattern formation in biofilms. Under planktonic growth conditions, several pathways have been shown to display high noise in gene expression in *B. subtilis*. These include competence[27], matrix production[28], sporulation[29] and, in some conditions, the general stress response[30]. However, it remains unclear how stochastic pulses in gene expression at the single cell level plays a role in the formation of spatial patterns in biofilms.

In this study, we use the *B. subtilis* general stress response pathway, mediated by the alternative sigma factor $\sigma^B$, as a model system to examine dynamic and heterogeneous gene expression in biofilms. $\sigma^B$ modulates more than 150 target genes in response to diverse stresses[31,32]. It is an ideal system for examining gene regulation dynamics in biofilms, as it is well characterised (see Fig. 1a for details of its regulation) and its gene regulatory dynamics have already been elucidated at the single cell level in microcolonies[30,33]. As a response to various stresses, entering the $\sigma^B$ active state can be considered an alternative protective strategy to that of forming a dormant spore (discussed in ref. [32]). Indeed, $\sigma^B$ activation has been shown to repress sporulation in planktonic growth[34–36].

$\sigma^B$ can be activated by two broad classes of stress: energy and environmental stress[32,37] (Fig. 1a). Energy stresses include ATP limitation through the addition of inhibitors (e.g. CCCP, MPA), entry into stationary phase, or carbon-limiting media. Environmental stresses shown to activate $\sigma^B$ include ethanol, NaCl, and heat. These responses have been characterised in planktonic growth, and it is unclear what the activation dynamics are of $\sigma^B$ during biofilm formation. Biofilms can contain spatially localised stress patterns[38], as well as gradients of nutrients away from the nutrient source[39], and thus to understand $\sigma^B$ activation during biofilm formation it is critical to examine gene expression in individual cells.

The energy stress pathway is mediated by a protein complex of the alpha/beta hydrolase RsbQ and serine phosphatase RsbP[40,41]. The environmental stress pathway is mediated through a sensing module termed the stressosome[42], which controls the activity of the phosphatase RsbU. Under unstressed conditions, $\sigma^B$ is inactive due to being bound to its anti-sigma factor RsbW. Under stress conditions, either RsbP or RsbU activates $\sigma^B$ by de-phosphorylating its anti-anti-sigma factor RsbV, allowing RsbV to bind the anti-sigma factor RsbW and release $\sigma^B$[43] (Fig. 1a). Although the regulation of the energy stress and environmental stress pathways are similar, strikingly different activation dynamics are observed in individual cells. Under energy stress conditions, $\sigma^B$ is activated in a series of stochastic pulses in microcolonies grown on agarose pads[30] due to continuous activation through the phosphatase RsbP. Environmental stresses however generate a single adaptive pulse in $\sigma^B$ activity[33] due to regulation of RsbU via the stressosome. A key aspect of the generation of these pulses is that $\sigma^B$ activates its own operon, consisting of $\sigma^B$, RsbV and RsbW. This results in both a positive feedback loop through the activation of $\sigma^B$, which amplifies fluctuations, and a negative feedback loop through the activation of RsbW, which terminates the pulse. Similar pulsing dynamics under energy stress have recently been observed in other alternative sigma factors that share a similar regulatory structure[44].

In this work, we examine $\sigma^B$ expression in an isolate of *B. subtilis* capable of forming biofilms. Under biofilm growth conditions we find that $\sigma^B$ is activated in a series of stochastic pulses, both in microcolonies and in developing biofilms. Moreover, we observe a gradient of $\sigma^B$ activation in *B. subtilis* biofilms, with the highest level of expression at the top of the biofilm. This gradient depends on the energy stress pathway. However, most spores are also found at the top of the biofilm[24], even though $\sigma^B$ has been shown to repress sporulation[34–36]. Through modelling and experiment, we propose that stochastic pulsing of $\sigma^B$ allows cells to activate either $\sigma^B$ or sporulation, allowing mutually exclusive cell states to co-exist in the same regions of the biofilm. Thus, stochastic pulses of gene expression can enable the formation of simple spatial patterns in biofilms.

## Results

### $\sigma^B$ pulses in microcolonies grown under biofilm conditions.

To investigate $\sigma^B$ activity in *B. subtilis* biofilms we constructed a $\sigma^B$ ($P_{sigB}$-YFP) reporter strain in a *B. subtilis* background capable of biofilm formation (NCIB 3610). The $P_{sigB}$-YFP construct contains the upstream region of RsbV from the $\sigma^B$ operon, which contains a $\sigma^B$-binding site. We used expression of this reporter as a proxy for $\sigma^B$ activity, as carried out previously[30]. This reporter strain also contained a constitutive $\sigma^A$ reporter for comparison ($P_{sigA}$-RFP), which contains the $\sigma^A$-binding site from the *trpE* gene (see Supplementary Methods for further details). Previous work using laboratory *B. subtilis* strains (Py79 and 168 Marburg backgrounds) incapable of forming biofilms demonstrated that $\sigma^B$ pulses stochastically under energy stress in individual cells growing in microcolonies[30]. However, recent work shows that media and growing conditions affect $\sigma^B$ dynamics[45]. We first checked whether pulsatile dynamics are observed using our reporter strain in microcolonies grown on agar pads of biofilm-promoting medium (MSgg, see the "Methods" section). The biofilm media was diluted 1:100 to promote microcolony formation, as carried out previously[24]. Under these conditions, pulses of gene expression were observed in the $P_{sigB}$-YFP reporter (Fig. 1b, d), but not in the constitutive reporter (Fig. 1c), similar to dynamics previously observed after application of energy stress[30].

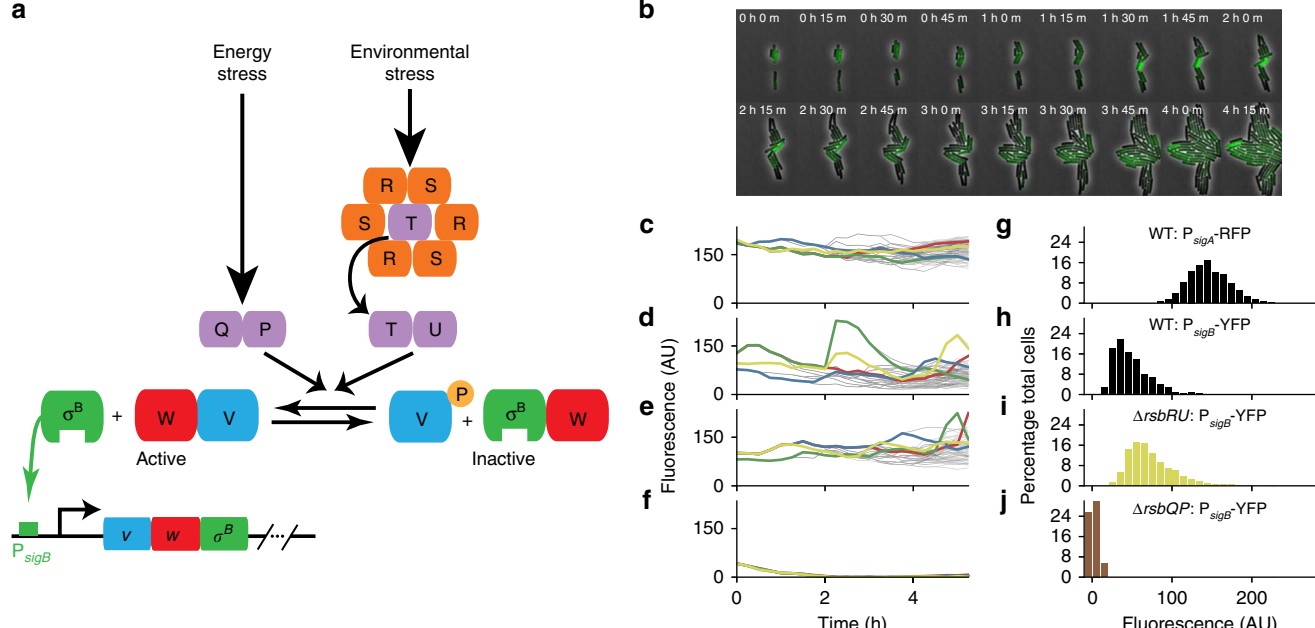

**Fig. 1 σ^B pulses in *B. subtilis* microcolonies via the energy stress pathway under biofilm conditions. a** Network diagram of σ^B circuit. Energy stress causes the complex RsbQP (QP, purple) to dephosphorylate RsbV (V, blue). Environmental stress releases RsbT (T, purple) from the stressosome (consisting of RsbR (R, orange) and RsbS (S, orange) subunits). RsbT then binds RsbU (U, purple) and activates it, with active RsbU dephosphorylating RsbV. Dephosphorylated RsbV (V, blue) binds to RsbW (W, red), releasing σ^B (σ^B, green) to bind to RNA polymerase and activate its target genes, including its own operon. **b** Filmstrip of σ^B activation in a WT *B. subtilis* microcolony grown in biofilm promoting medium. Heterogeneous expression levels of P_*sigB*-YFP (green) reflect pulsing activity. Numbers indicate time in hours and minutes. **c** Single cell fluorescence values for constitutive fluorescent reporter (P_*sigA*-RFP) over time (representative traces). **d–f** Single cell fluorescence values for P_*sigB*-YFP reporter over time for WT **d**, Δ*rsbRU* **e**, and Δ*rsbQP* **f**, strains. The time in hours is relative to the start of analysis. **g, h** Histograms of fluorescence intensities of WT cells for the constitutive reporter P_*sigA*-RFP **g** and P_*sigB*-YFP reporter **h** (For both, cells: 1157, movies: 6, experiments 2). **i, j** Histograms of P_*sigB*-YFP expression for Δ*rsbRU* strain **i** (cells: 1178, movies: 5, experiments: 3) and Δ*rsbQP* strain **j** (cells: 1290, movies: 8, experiments 3). Histograms in **g–j** include all frames in movies with more than 8 cells. Source data are provided as a Source Data file.

The observed pulsed σ^B gene expression is not dependent on the environmental stress pathway, as pulsing gene expression is still observed after knocking out the operon containing the main environmental stress pathway components RsbR, RsbS, RsbT, and RbsU (Δ*rsbRU*) (Fig. 1e). The energy stress pathway, however, is required for the pulsing dynamics, as P_*sigB*-YFP levels drop to background levels in a reporter strain where RsbQ and RsbP are knocked out (Δ*rsbQP*) (Fig. 1f, j). The pulsed gene expression results in a long tailed distribution of fluorescent reporter intensities in snapshots of gene expression from individual cells in the WT (mean 49.06 ± 16.14, coefficient of variation (CV) 0.42, Fig. 1h) and Δ*rsbRU* backgrounds (mean 72.70 ± 19.07, CV 0.27, Fig. 1i). The long tailed distribution is caused by σ^B infrequently pulsing on to a high level. This results in most cells having low P_*sigB*-YFP expression levels, with only a few cells having high expression levels at a given time. This causes the σ^B pathway to have a higher level of noise in gene expression than the constitutive P_*sigA*-RFP reporter, which has a lower CV (CV 0.13, Fig. 1g).

**σ^B forms an activation gradient in biofilms**. We next examined σ^B expression in developing biofilms grown on a solid agar surface (see the "Methods" section). To quantify σ^B regulation in the biofilm, we carried out confocal microscopy on thin slices of samples from the centre of biofilms at 24, 48, 72, and 96 h after inoculation. (In this paper our samples are from the centre of the biofilm colony, which has a wrinkle morphology distinct from the smoother edge of the biofilm (Supplementary Fig. 1).) The microscopy settings were configured to minimise possible bleed-

through of signal between the acquisition channels (see the "Methods" section). We observed a distinct pattern of σ^B activation, with a gradient of σ^B expression that peaks at the top of the biofilm (Fig. 2a, d). The gradient can be observed at multiple timepoints during biofilm development (see Supplementary Fig. 2A). This gradient in expression was not due to variation in maturation properties of the yellow fluorescent protein through the biofilm, as the expression of a reporter using the same fluorescent protein for a different alternative sigma factor, σ^W (P_*ydbS*-YFP) that is active during biofilm formation[46–48] (see Supplementary Fig. 3), and P_*sigA*-YFP (see Supplementary Fig. 4) were uniform throughout the biofilm and did not display a gradient of expression.

Recent work has found that *B. subtilis* often forms an association with plant roots in the soil to form a possible symbiotic relationship with the plant[49]. In order to test whether a gradient in σ^B expression is a general phenomenon during biofilm development, we examined our *B. subtilis* P_*sigB*-YFP reporter cells growing on *Arabidopsis thaliana* roots. We observed a gradient in P_*sigB*-YFP expression in WT cells that was not present in a Δσ^B background (see Supplementary Fig. 5), suggesting that a σ^B gradient can be present in a range of growth conditions.

We next examined which of the σ^B major input pathways (energy or environmental stress) are required for the generation of the σ^B expression gradient in biofilms. First, to test the role of the environmental stress pathway we examined σ^B expression in the Δ*rsbRU* reporter strain using our standard biofilm colony growing condition on MSgg agar. The overall pattern of σ^B activation in the Δ*rsbRU* background resembled WT (Fig. 2b, d, Supplementary Fig. 2B). In contrast, deletion of the energy stress

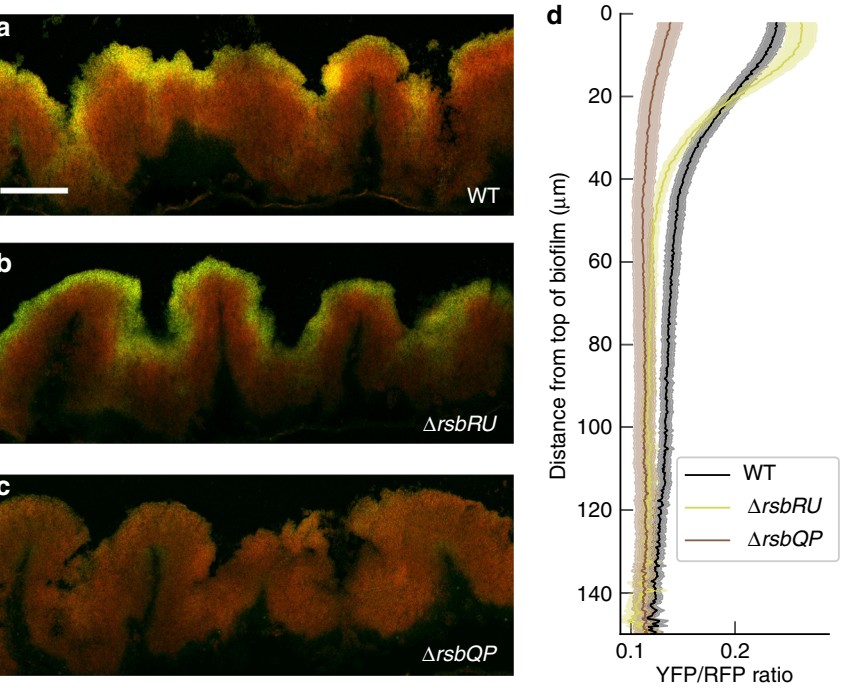

**Fig. 2 The energy stress pathway mediates a $\sigma^B$ expression gradient from the top of the biofilm. a–c** Confocal microscopy images of slices of central parts of 48 h biofilm colony with merged channels for constitutive fluorescent reporter, P$_{sigA}$-RFP, in red and P$_{sigB}$-YFP in green. **a** WT cells, **b** ΔrsbRU cells, **c** ΔrsbQP cells. Scale bar 100 μm. **d** The ratio of P$_{sigB}$-YFP to P$_{sigA}$-RFP expression levels in the three strains as measured from the top of the colony, error bars indicate the SEM (WT: n = 13 images from three experiments, ΔrsbRU: n = 10 from two experiments, ΔrsbQP: n = 11 from two experiments). Source data are provided as a Source Data file.

pathway (ΔrsbQP) significantly reduced $\sigma^B$ expression in biofilms and resulted in an attenuated gradient (Fig. 2c, d, Supplementary Fig. 2C).

To test whether $\sigma^B$ might pulse in biofilms, we examined whether a long-tailed distribution of P$_{sigB}$-YFP fluorescence intensities was present in snapshot images of biofilms, similar to the long tails observed in small colonies that exhibited pulsing (Fig. 1h, i). Segmentation of individual cells in biofilm slices from high magnification images revealed variation in intensity of P$_{sigB}$-YFP expression in adjacent cells. At the top edge of the biofilm we observed a long tail in the histogram of cell expression, again dependent on the energy stress pathway (Fig. 3). When we sampled histograms throughout the remainder of the biofilm we observed that the distribution became narrower and less long tailed as we moved away from the top of the biofilm (Supplementary Fig. 6), due to the diminished activity of $\sigma^B$.

**$\sigma^B$ pulses occur throughout biofilm formation**. We hypothesised that the observed gradient in $\sigma^B$ levels (Fig. 2) was generated by $\sigma^B$ pulsing in a growing biofilm. To confirm this we developed a time-lapse protocol that allowed us to examine single cell $\sigma^B$ activation dynamics across a part of a growing biofilm colony (see the "Methods" section). By growing the biofilm against the glass bottom of a dish (Fig 4a), we were able to image, with single cell resolution, the growth of a biofilm that exhibited characteristic biofilm properties such as wrinkle formation (Supplementary Fig. 1 and Supplementary Movie 1). We imaged the ΔrsbRU background to avoid $\sigma^B$ activation due to repeated laser illumination through the environmental stress pathway, as the pathway can be activated through the blue light sensor, ytvA[50].

We were able to observe the establishment of the P$_{sigB}$-YFP gradient (Fig. 4b, c) that was present in snapshots from cryoslices of the biofilm (Supplementary Fig. 2). The P$_{sigB}$-YFP gradient developed through time, peaking at 72 h after inoculation. We

then examined $\sigma^B$ dynamics in individual cells at a high frequency, by taking time-lapse images every 10 min for up to 96 h. By extracting single cell time-traces during the development of the biofilm, we found sustained pulsing of $\sigma^B$ in individual cells (Fig. 4d) at the top of the biofilm throughout biofilm development (Supplementary Movie 2). These pulses lead to a long-tailed histogram of single cell P$_{sigB}$-YFP levels (Supplementary Fig. 7D), similar to that observed in snapshots of biofilm slices (Fig. 3), which was absent from the P$_{sigA}$-RFP control (Supplementary Fig. 7B). We also observed heterogeneous expression, as well as a gradient of gene expression across the biofilm, for reporters of two genes regulated by $\sigma^B$, csbB[51], and yflA[52] (Supplementary Fig. 8). Thus, $\sigma^B$ pulses through the energy stress pathway during biofilm development, resulting in a heterogeneous distribution of $\sigma^B$ active and inactive states, which are reflected in the expression levels of $\sigma^B$ targets.

**Pulsing allows mutually exclusive states to co-exist**. As well as the gradient of $\sigma^B$ expression, we also observed a gradient of spore formation (using a reporter for late stage sporulation (P$_{sspB}$-YFP),[53]) in the biofilm, with the highest number of cells transitioning to spores at the top of the biofilm (blue line, Fig. 5a). However, $\sigma^B$ has been proposed to repress sporulation in planktonic growth[34–36]. The repression occurs due to $\sigma^B$ activating expression of the phosphatase Spo0E, a negative regulator of the master regulator of sporulation, Spo0A[34,35]. We first confirmed that sporulation is not required for the gradient in $\sigma^B$ expression (Supplementary Fig. 8D) by deleting $\sigma^F$, which is required for spore formation. Next, to test whether $\sigma^B$ represses sporulation during biofilm formation we constructed a strain with reporters for both $\sigma^B$, P$_{sigB}$-YFP, and an early sporulation gene—spoIID (P$_{spoIID}$-CFP)[54]. CFP illumination caused significant phototoxicity during time-lapse imaging of developing biofilms, so we reverted to examining snapshots of expression from biofilm

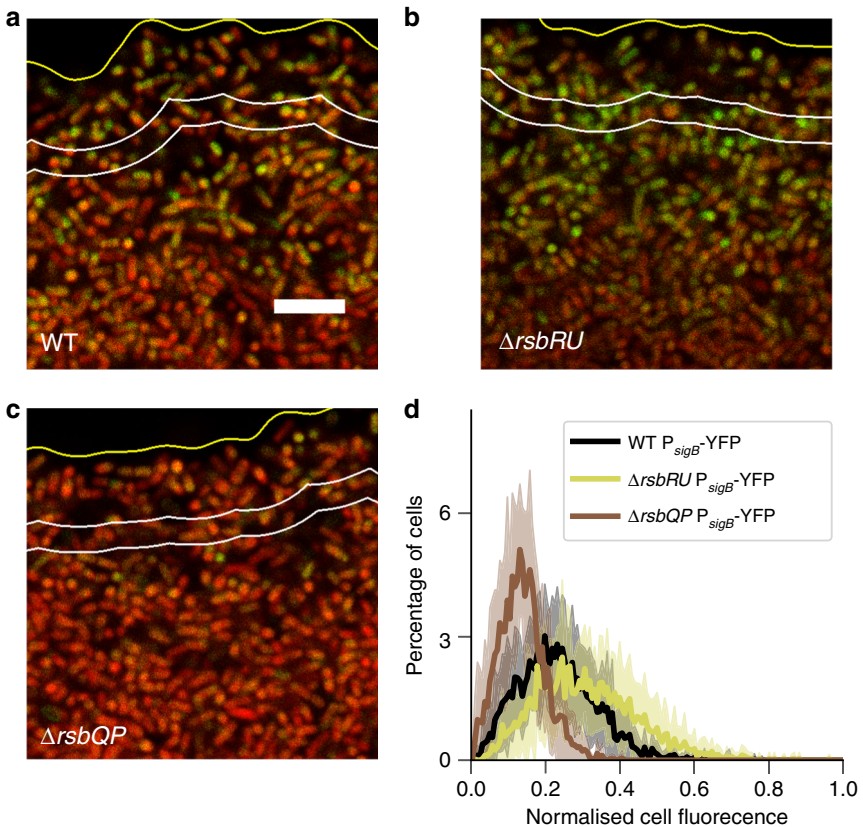

**Fig. 3 Heterogeneous activation of P$_{sigB}$-YFP in individual cells in biofilm colonies is mediated through the energy stress pathway. a–c** High magnification confocal images of the top 25 μm of the central part of a *B. subtilis* biofilm. Merged P$_{sigA}$-RFP (red) and P$_{sigB}$-YFP (green) channels for **a** WT, **b** *ΔrsbRU*, and **c** *ΔrsbQP,* respectively. Yellow lines indicate the top edge of the biofilm and the white lines mark the 5 and 7 μm isolines. Scale bar is 5 μm. **d** Histograms of normalised cell fluorescence in a range from 5 to 7 μm from the top of the biofilm. The YFP values are normalised by the mean RFP fluorescence in the range for that image. The error represents the standard deviation of the histogram for each image (WT histograms from 4231 cells, $n$ = 24 images from five experiments, *ΔrsbRU* histograms from 3268 cells, $n$ = 18 from three experiments, *ΔrsbQP* histograms from 2481 cells, $n$ = 16 from three experiments). Source data are provided as a Source Data file.

slices. We found that $\sigma^B$ and sporulation activity are anti-correlated at the cellular level (Fig. 5b, c), with cells that have activated $\sigma^B$ to high levels having lower levels of P$_{spoIID}$-CFP expression (best fit slope of −0.31). This was not apparent if the values of P$_{spoIID}$-CFP and P$_{sigB}$-YFP were randomly shuffled (best fit slope of 0) (Supplementary Fig. 9), confirming that the pathways are anticorrelated rather than just not correlated.

Given the anti-correlation between cells entering sporulation and cells activating the general stress response (see Supplementary Fig. 9), we hypothesised that the pulsed expression of $\sigma^B$ allows the coexistence of both transcriptional states at the top of the biofilm (Fig. 5a, b). Due to the long-tailed distribution of $\sigma^B$ expression values (Fig. 3), only a fraction of cells exist in the high $\sigma^B$ and low P$_{spoIID}$-CFP expression state (Fig. 5) at any one time. This suggests that pulsed $\sigma^B$ expression allows a fraction of cells to enter the high $\sigma^B$ state without shutting off sporulation in all cells, which would occur if $\sigma^B$ was expressed in a non-pulsatile manner to a high level in all cells at the top of the biofilm. Deleting $\sigma^B$ resulted in a qualitatively similar sporulation gradient as compared to WT, confirming that the pulsing of $\sigma^B$ allows a proportion of cells to have levels of $\sigma^B$ that repress sporulation without interfering with the overall pattern of sporulation in the biofilm (Supplementary Fig. 10). To investigate further the functional role of pulsed gene expression in biofilm pattern formation, we developed a simple stochastic model of two pulsing regulators for sporulation ($A$) and $\sigma^B$ ($B$), with one ($B$) able to block transcription of the other ($A$), with $A$ leading to sporulation

(Fig. 6a). Both pathways were assumed to pulse, as the sporulation pathway has also been observed to pulse in microcolonies grown on agarose pads[55–58].

We tested how the pulsed activation of two competing pathways affected cell fate in our model, and how these effects could be modulated by spatial signals. In our model, the frequency of stochastic pulses of activity is determined by a gradient representing "stress", where the pulse frequencies of $A$ and $B$ are scaled independently from each other along this gradient. The finite duration of the pulses allows each system the opportunity to be dominant for a period of time. We observed a gradient of both $A$ and $B$ species that rose up the stress gradient (Fig. 6c, d), as observed experimentally for sporulation and for the general stress response (Fig. 5a). We then tested the effects of modulating pulse frequency of the repressor $B$ on the sporulation gradient. Doubling the pulse frequency of $B$ at all points along the gradient caused the peak of sporulation to be shifted away from the top (Fig. 6c), due to the increased repression of the sporulation pathway.

We tested the model prediction that increasing $\sigma^B$ activity should alter the gradient of spores in the biofilm by inserting an additional copy of the genes (*rsbQ-rsbP*) controlling the energy stress pathway (2 × *rsbQP*). We did this as increasing phosphatase levels was previously shown to increase $\sigma^B$ pulsing activity in microcolonies[30]. We made two such strains, one with a reporter for $\sigma^B$ (P$_{sigB}$-YFP), and another with a reporter for late stage sporulation (P$_{sspB}$-YFP) (see the "Methods" section). We examined the effect of increasing RsbQP copy number on the $\sigma^B$

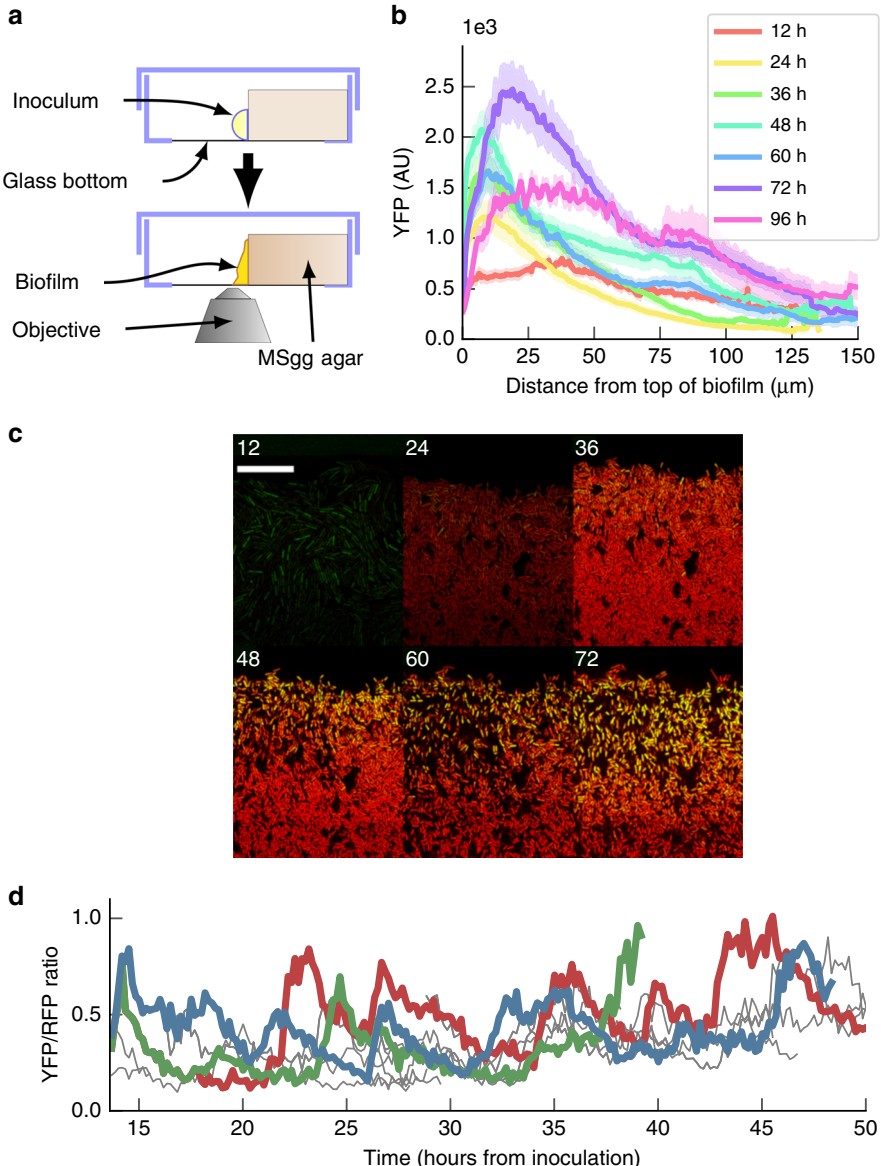

**Fig. 4 $\sigma^B$ pulses in individual cells during biofilm development. a** Experimental setup used to capture time-lapse confocal images of the growing biofilm. A glass bottom dish is partially filled with MSgg agar. Cells are inoculated on the agar surface in contact with the glass. The biofilm grows on the agar surface, allowing time-lapse microscopy across a cross-section of the biofilm. **b** Quantification of $\sigma^B$ gradient from live biofilm snapshots taken every 12 h, error bars indicate the SEM. $n = 17$ from two experiments. **c** Snapshots of the top of the live biofilms taken at 12 h intervals after inoculation shows development of a $\sigma^B$ gradient of expression. Snapshots are merged $P_{sigA}$-RFP (red) and $P_{sigB}$-YFP (green) channels. Scale bar is 20 μm. **d** Ratiometric expression of $P_{sigB}$-YFP/ $P_{sigA}$–RFP in a selection of cells tracked for over 40 h shows sustained pulsing of $P_{sigB}$-YFP during biofilm growth. Source data are provided as a Source Data file.

gradient, and on the distribution of spore forming locations. As previously observed in liquid culture[30], we observed higher single cell $\sigma^B$ expression in the $2 \times rsbQP$ strain than WT, with the single cell distributions remaining heterogeneous, characteristic of pulsing. Pulsing could also still be observed in timelapse movies of a $\Delta rsbRU$ $2 \times rsbQP$ strain (Supplementary Movie 3). We then addressed the effects of the higher $\sigma^B$ activity during biofilm formation. The $2 \times rsbQP$ strain has a much stronger $\sigma^B$ gradient than WT (Fig. 7b and Supplementary Fig. 11A) and more extreme heterogeneity in $\sigma^B$ expression (Supplementary Fig. 11B). In $2 \times rsbQP$ the spore forming region is shifted away from the top of the biofilm (Fig. 7a, d) compared to WT (Fig. 7a, c), confirming that $\sigma^B$ is repressing sporulation in biofilms, and that pulsing of $\sigma^B$ in WT cells allows the coexistence of both cell states.

## Discussion

Our quantitative single cell microscopy has revealed how *B. subtilis* can use dynamic gene expression to generate spatial patterns of cell states. Using a fluorescent reporter for $\sigma^B$ expression we found that $\sigma^B$ is expressed heterogeneously throughout biofilm development (Fig. 3) and by tracking single cells during biofilm development we observed that this heterogeneous expression is generated by the pulsing on and off of $\sigma^B$ (Figs. 4 and 1). $\sigma^B$ activity, dependent on the energy stress pathway, is present as a gradient, with the highest levels at the top of the biofilm. We also found a gradient of spore density that is similar to the $\sigma^B$ gradient (Fig. 5b), even though sporulation is repressed by $\sigma^B$ activity. Modelling and experiment suggest that pulsed activation of $\sigma^B$ allows these two mutually exclusive states

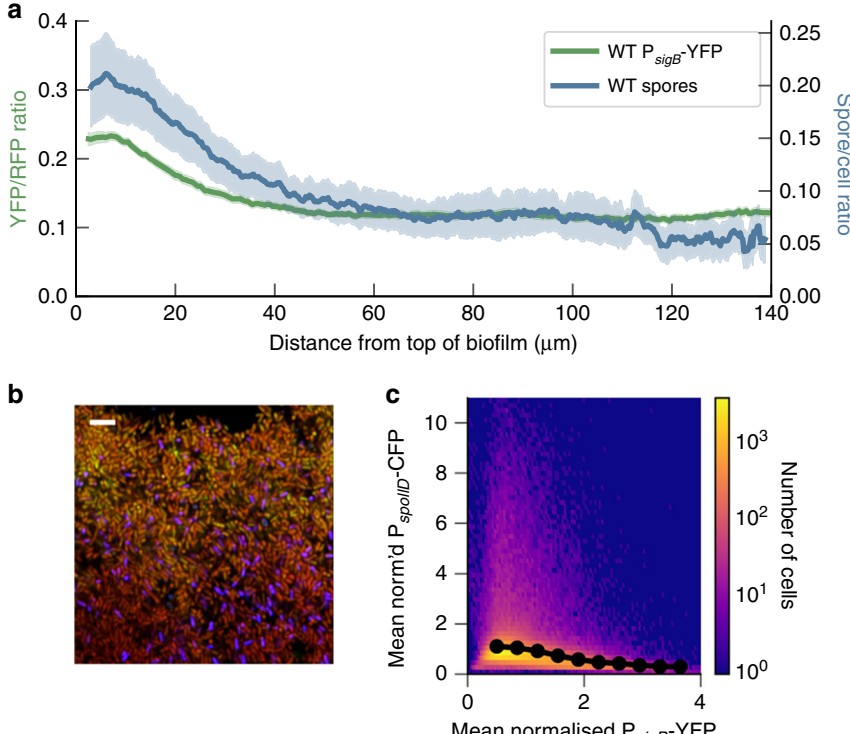

**Fig. 5 *B. subtilis* biofilms exhibit co-located spore and $\sigma^B$ gradients, even though $\sigma^B$ represses sporulation in single cells. a** The wild type gradients of $\sigma^B$ (the ratio of $P_{sigB}$-YFP to $P_{sigA}$-RFP, $n = 24$ from five experiments) and spore density (calculated from cells expressing late stage sporulation marker, $P_{sspB}$-YFP, $n = 11$ from four experiments). Error bars indicate the SEM. **b** Representative image showing coexistence of sporulating cells and $\sigma^B$ expressing cells at the top of the biofilm. Green channel is $P_{sigB}$-YFP, blue channel is $P_{spoIID}$-CFP, red channel is $P_{sigA}$-RFP. The top of the biofilm is at the top of the image. Scale bar is 5 μm. **c** Bivariate histogram showing the frequency of cells co-expressing $P_{sigB}$-YFP and $P_{spoIID}$-CFP. The fluorescent values are mean normalised for each image. The data represent two tilescan images, each covering ~3 mm of biofilm, from two different experiments and 372,689 cells. Source data are provided as a Source Data file.

to exist in the same layer of the biofilm, pointing to a function for noisy gene regulation in biofilm formation.

Gradients of nutrients[59] and oxygen[22,60] in biofilms have been observed in many species of bacteria[61], which can lead to gradients in gene expression[38,59,62,63]. We observe a gradient of $\sigma^B$ activation, dependent on the energy stress pathway, at the top of the biofilm that could correspond to a gradient of nutrients that depletes near the top of the biofilm. Interestingly the general stress response regulator *rpoS* is activated only in the top half of *E. coli* biofilms[38], and in *Pseudomonas aeruginosa*, heterogeneous levels of *rpoS* mRNA have been recorded in the top layers of the biofilm[64]. In future it will be important to see, at the single cell level, whether these pathways display significant heterogeneity, and whether a similar functional role for this heterogeneity, allowing multiple cell states to exist in the same layer of the biofilm, is apparent.

We were able to observe single cell $\sigma^B$ dynamics both in microcolonies (Fig. 1) and in developing biofilms (Fig. 4). Our approach is complementary to recent single cell time-lapse analysis of bacterial biofilms based on microfluidic devices[65–67]. Microfluidic devices have advantages for examination of biofilm formation, including allowing constant environmental conditions and the easy manipulation of conditions. However, biofilms grown in these devices do not exhibit the full range of physiological phenotypes exhibited by biofilms grown on solid surfaces. Our novel single-cell time-lapse protocol allows gene expression dynamics to be tracked during biofilm growth on solid surfaces, and can be readily applied to other microbial pathways.

We were able to examine the dynamic interactions between $\sigma^B$ and sporulation in a simple mathematical model (Fig. 6). The

model qualitatively captures how pulsing of a repressor, in combination with a gradient of signal, can generate a simple spatial pattern. Going forward, it will be interesting to use such a model to test the differences between unidirectional coupling and a mutually repressive bistable switch circuit[68]. However, our model does not account for the complex interactions of biofilm growth, spatial signals, and the detailed mechanisms of the regulation of $\sigma^B$ or sporulation. In future this conceptual model should be expanded to develop a mechanistic understanding of the interactions between $\sigma^B$ and sporulation in biofilm formation, which can capture more detailed aspects of their interactions. It will also be important to examine how the pattern observed here can contribute to the fitness of the colony. For example, the top of the biofilm consisting of both a protected layer of $\sigma^B$ active cells and spores could be important for efficient spore dispersal.

In computation, randomness or noise can be harnessed to compute a function more quickly or with a simpler algorithm[69–71] than is possible in a deterministic setting. Here we observe noise operating at the single cell level to control the distribution of cell types. It will be interesting to test what range of patterns noisy systems can generate compared to deterministic systems and whether there is a fundamental advantage to noise in development.

Multiple regulatory pathways have been shown to be expressed heterogeneously in *B. subtilis* under planktonic growth conditions. In addition to the general stress response[30] and sporulation[29], these include competence[27], motility, and matrix production[28]. Pulsing in several *B. subtilis* alternative sigma factors has also been observed under energy stress, potentially allowing time-sharing of RNA polymerase during planktonic

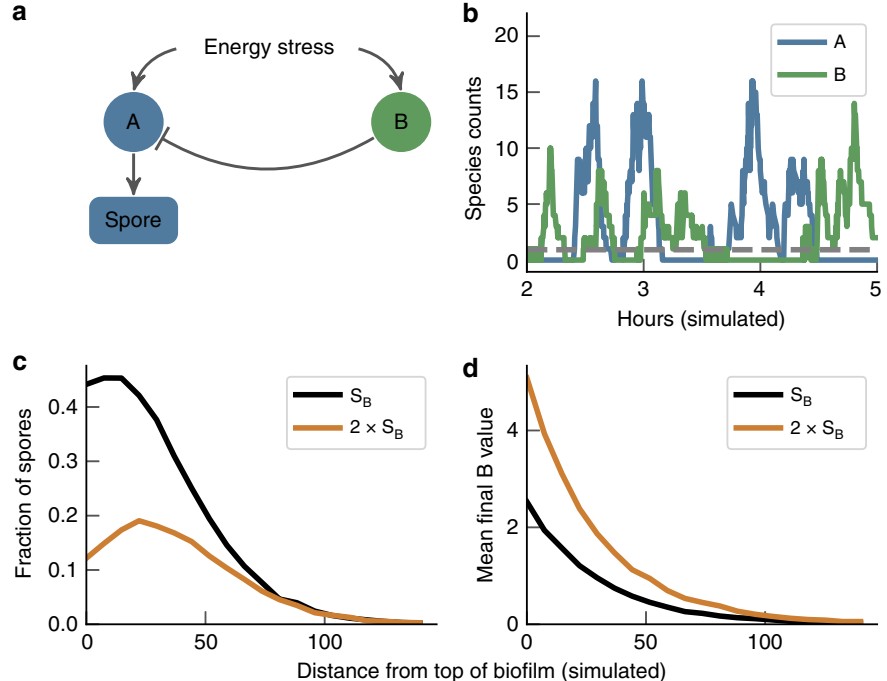

**Fig. 6 A simple model predicts suppression of sporulation at the top of the biofilm with increased pulse frequency of $\sigma^B$. a** Model network diagram. **b** Example traces of species **a** and **b** exhibiting pulses at the top (distance 0) of the simulated biofilm. The grey dashed line is the spore threshold (set at 1 for these parameters. A cell must express A for more than 30 simulated minutes above the threshold value to become a spore. For details see the "Methods" section). **c** Fraction of simulations where the value of species A exceeded the spore threshold for more than 30 simulated minutes where the B pulse frequency parameter $s_B = 0.25$ and with the pulse frequency doubled $2 \times s_B = 0.5$. The A pulse frequency parameter is set to $s_A = 0.7$. **d** Shows the mean final value of species B, recorded from 10,000 simulations for $s_B = 0.25$ and $2 \times s_B = 0.5$. Source data are provided as a Source Data file.

growth[44]. In future it will be important to test whether such noisy gene regulation in other pathways plays a role in biofilm formation.

## Methods

**Bacterial strains, media, and culture conditions**. Biofilm forming *B. subtilis* strains are derived from NCIB3610 obtained from the Bacillus Genetic Center. Donor strains for reporter constructs are derivatives of 168 Marburg. The complete strain list is provided in Supplementary Table 1. Genomic integration of reporter sequences was achieved using the pLK plasmid as described in ref. [72]. Briefly, the pLK plasmid (which encodes an IPTG inducible competence regulon) was introduced to the NCIB3610 strain by protoplast transformation. ComK expression was then induced in NCIB3610+pLK by 1 mM IPTG in the presence of the integrative plasmid vector or genomic DNA. Target integration was confirmed by antibiotic selection and PCR. The pLK plasmid is unstable at high temperatures and was cured by incubating the resulting strains at 50 °C for 8 h.

Alternatively reporter genes were transferred to NCIB3610 using SPP1 phage-mediated transduction[73]. Briefly, 200 µL of overnight culture of donor bacterial strain was added and incubated with phages in 200 µL TY (LB plus 10 mM MgSO$_4$ and 1 µM MnSO$_4$) for 15 min at 37 °C without shaking. Then cells with phage were mixed with 3 mL of melted top TY agar and poured onto a fresh TY plate and incubated until confluent lysis developed. 5 ml TY was added to each plate and the top agar layer containing phages was scrapped, collected, and centrifuged. Supernatant with phages was collected and filtered. Phages were stored at 4 °C. Recipient cells were grown overnight in TY at 37 °C and incubated with phages containing donor DNA for 30 min at 37 °C, media was removed by centrifugation, cells resuspended in TY and plated on TY plates with 10 mM sodium citrate and selective antibiotics. After O/N growth another round of antibiotic selection was performed and selected colonies were checked for proper DNA integrations.

Finally, we also transformed reporter strains into the easily transformable NCIB3610 background from the Kearns laboratory, which is identical to NCIB3610 except for a single point mutation, comI(Q12L) in an endogeneous plasmid that increases natural competence[74]. For detailed information regarding the construction of strains and plasmids used in this study, please see the Supplementary Methods.

For routine growth, cells were propagated in LB medium. When necessary, antibiotics were used at the following concentrations: spectinomycin (100 µg mL$^{-1}$), chloramphenicol (5 µg mL$^{-1}$), kanamycin (10 µg mL$^{-1}$), tetracycline (10 µg mL$^{-1}$), and erythromycin (1.25 µg mL$^{-1}$). For biofilm growth we used MSgg medium[23,75] (5 mM potassium phosphate (pH = 7.0), 100 mM 3-(N-morpholino)

propanesulfonic acid (pH = 7.0), 2 mM MgCl$_2$, 700 µM CaCl$_2$, 50 µM MnCl$_2$, 50 µM FeCl$_3$, 1 µM ZnCl$_2$, 0.5% glycerol, 2 µM thiamine–HCl, 0.5% potassium glutamate) fortified with 1.5% bacto agar (Fisher Scientific).

***B. subtilis* biofilm growth on *A. thaliana* roots assay**. The experiments to show *B. subtilis* biofilm formation on plant roots were performed as described in ref. [49]. Briefly, *A. thaliana* Col-0 seedlings were grown on $\frac{1}{2}$ Murashige–Skoog medium (0.22% MS salts + vitamins (Duchefa M0222) supplemented with 0.8% plant agar, pH = 5.7–5.8) at 20 °C. Plants were grown in a 16:8 h light:dark cycle growth chamber for 6 days. Next, the seedlings were transferred to 300 µl of MSNg medium (5 mM potassium phosphate, pH 7; 0.1 M MOPS, pH 7; 2 mM MgCl$_2$; 0.05 mM MnCl$_2$; 1 mM ZnCl$_2$; 2 µM thiamine HCl; 700 µM CaCl$_2$; 0.2% NH$_4$Cl supplemented with 0.05% glycerol) in 48-well plates. Medium with plants was inoculated with *B. subtilis* cells at OD 600 = 0.02. Plates with plants and bacteria were put on an orbital shaker at 100 rpm and incubated in the same plant growth chamber for 48 h. Plant roots were subjected to confocal microscopy directly after that.

**Biofilm growth, sample collection, and thin sectioning**. To prepare thin slices for imaging, biofilm colonies were grown as follows: cells were scraped from an overnight LB plate and re-suspended in LB medium to $A_{600} = 1.0$ optical units. 2 µL of cell suspension was put on the surface of an MSgg agar plate. Biofilm colonies were collected at 24, 36, 48, 72, and 96 h after incubation at 30 °C and flash-frozen in liquid nitrogen overlaid with Tissue-Tec O.C.T. compound (Sakura Finetek, ref 4583) in custom foil moulds and kept at −80 °C. 8 µm thin slices were made using a Leica CM3050S cryomicrotome, collected on SuperFrost Plus microscope slides (VWR cat. no. 631-0108) and stored at −80 °C before use. For imaging, biofilm slices were overlaid with the mounting medium (0.5% N-propyl gallate, 50% glycerol and PBS (pH 7.4))[24].

**Agarose pad time-lapse microscopy**. Time-lapse imaging of single bacterial cells was performed as described in refs. [30,76]. Cells were grown overnight on LB plates with appropriate antibiotics. Then cells were diluted to $A_{600} = 0.01$ in MSgg and 2 µL of this suspension was spotted on low melting temperature agarose pads made with 1:100 MSgg diluted in water. Cells were imaged with a Nikon Ti-E inverted microscope using a CFI Plan Apo DM Lambda ×100 oil objective (N.A. = 1.45), a Photometrics Coolsnap HQ2 camera, and using an automated time-lapse imaging platform. Time-lapse images were taken every 10 min for 10 h and were processed and analysed using Schnitzcells[76]. Movies were not analysed if they were out of

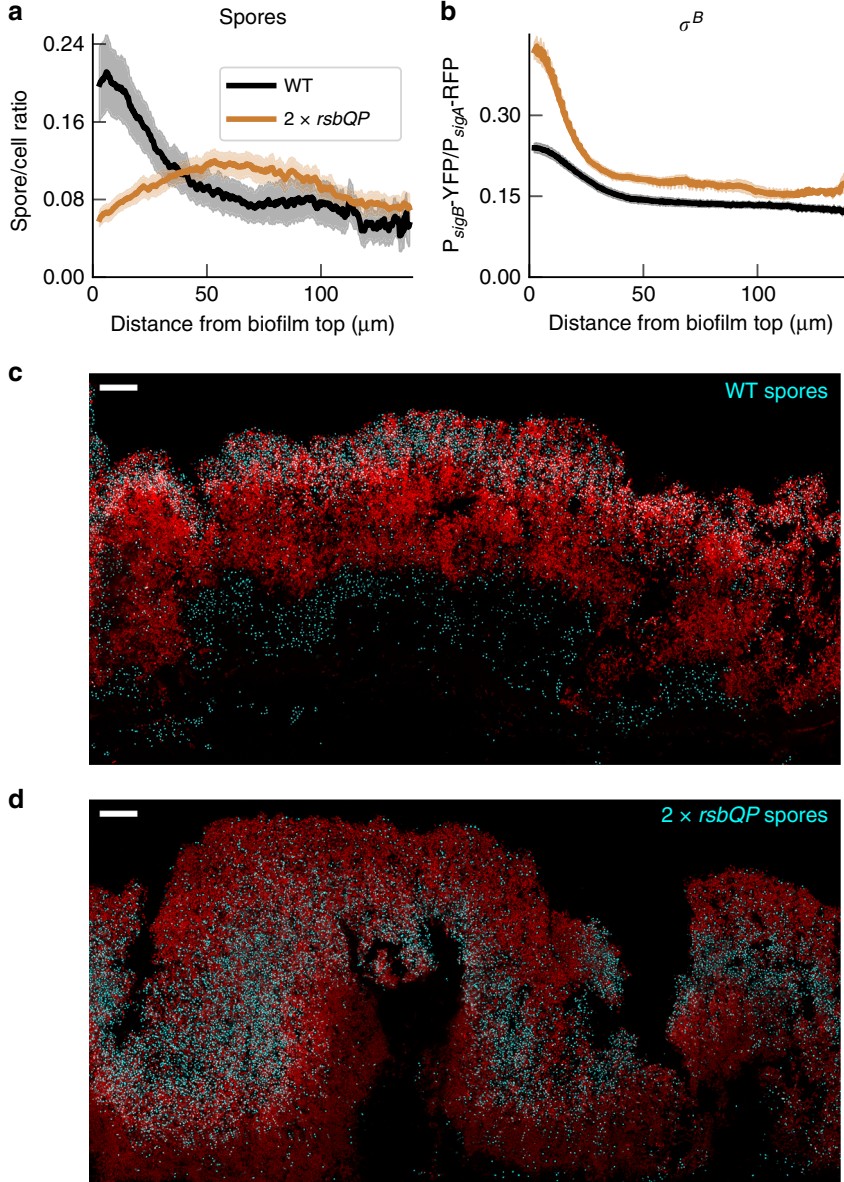

**Fig. 7 Distribution of sporulating cells in biofilm colony is modulated by $\sigma^B$ expression. a** Spore density at the top of biofilm colonies in WT and 2 × *rsbQP* strains. The graph represents a ratio of amount of cells identified as spores (by expression of the $P_{sspB}$-YFP fluorescent marker) to the combined amount of all cells expressing $P_{sigA}$-RFP and spores (see the section "Methods" for details of quantification). For WT, data is replotted from Fig. 5a, $n = 11$ from four experiments, for 2 × *rsbQP* data, $n = 12$ from four experiments. Error bars represent SEM. **b** Doubling of *rsbQP* gene copy number increases and sharpens $\sigma^B$ expression gradient in biofilm colonies. The graph represents a ratio of $P_{sigB}$-YFP to $P_{sigA}$-RFP expression in WT (black) and 2 × *rsbQP* (brown) backgrounds in biofilm colonies as measured from the distance from the colony top. (WT data $n = 13$ from three experiments, 2 × *rsbQP* $n = 7$ from 1 experiment). Error bars show the SEM. Scale bars are 25 µm. **c, d** representative images of the spore pattern in a WT and 2 × *rsbQP* biofilm respectively, $P_{sigA}$-RFP in red and $P_{sspB}$-YFP in cyan. All samples are from the central part of the biofilm and collected at 48 h after inoculation. Note that in **c**, between the biofilm and agar, is a layer of spores. This is typical of WT biofilms and is not reflected in the spore gradient **a** since there are no cells to compute a ratio. The full context for **c** and **d** is shown in Supplementary Fig. 12. Source data are provided as a Source Data file.

focus, if the cells piled vertically too early, or if cells grew too slowly. Individual cells that behaved unusually, (e.g. stopping growth or lysing) were excluded from the analysis. When analysing single cell data we discard data from the first frames of the movie (before 8 cells) to allow cells to adjust to growth on the agarose pads and to avoid inclusion of cells with fluorescent protein levels that could reflect the previous plankonic growth or the transition to the agarose pad.

**Thin slice imaging**. Imaging of biofilm slices (Fig. 2) with fluorescent reporters was performed using a Zeiss LSM780 laser scanning confocal microscope with a dry 10x Plan Apochromat (NA 0.45). Images were acquired using sequential imaging and saved as 16 bit images with 1 pixel representing 0.59 µm. For single cell imaging with three fluorescent reporters (Fig. 5, Supplementary Fig. 9) we used an oil immersion ×63 Plan Apochromat (NA 1.4) objective. We used the LSM780

built-in linear unmixing feature of the confocal microscope to reduce bleed-through of signal.

For single cell imaging of biofilm slices (Figs. 3, 11 and Supplementary Figs. 5, 6) we used a Zeiss LSM700 with an oil immersion ×63 Plan Apochromat (NA 1.4) objective. Images were acquired using sequential imaging with a 488 nm laser at 2% power and a 555 nm laser at 2% power. The pinhole was 47 µm for both channels. Images were stored as 16 bit images with 1 pixel representing 0.05 µm. The beam splitter for the yellow channel was MBS 405/488/555/639 DBS1: 550 nm, and for the red channel MBS 405/488/555/639 DBS1: 600 nm. To quantify sporulation (Fig. 7, Supplementary Figs. 10 and 12) we made long tilescans of biofilm slices. Regions of biofilm were excluded from analysis if they were physically damaged during sectioning. We also excluded regions where the biofilm folded over on itself or where the top was not easily discernible.

Images were analysed using custom software. Due to decreasing cell density there is ambiguity as to where the biofilm starts at the air interface, this interferes with spore density calculations so we discard the top 2 μm of all biofilm gradient data.

The LSM700 was also used to capture the live biofilm images with single cell resolution in Supplementary Figs. 8 and 4 using the above settings. For the live biofilm gradients in Supplementary Fig. 8 we used the LSM700 with the same settings but with a dry ×20 Plan Apochromat (NA 0.8) objective with the pinhole for the 555 nm laser at 33.40 μm and for the 488 nm laser the pinhole was at 34.52 μm. Images were acquired using sequential mode and saved as 16 bit images with 1 pixel covering 0.31 μm.

Images were discarded if the live cells moved between sequential red/yellow laser scans. Images were also excluded from analysis if they contained very few cells or were mostly spores rather than live cells. The cells from this dataset (Supplementary Figs. 4 and 8) exhibited a bimodal pattern of red channel (constitutive $P_{sigA}$-RFP) intensity. We concluded, on visual inspection, that the fainter population were cells in a different focal plane. We exclude these out of plane cells by only considering cells with mean cell red intensity >6800. We chose this value by fitting two Guassian distributions to the bimodal populations, the threshold value is the mean minus the standard deviation of the brighter population of cells.

**Imaging of live biofilm development.** To grow colonies suitable for live time lapse confocal imaging we first filled 50 mm glass bottomed dishes (Wilco Wells HBST-5040) with 9 mL of MSgg agar (described above). The dishes were left closed over night at room temperature, then stored at 4 °C. Before inoculation, we cut the agar into two segments, dividing the agar into approximately one-third and two-thirds. The cut was made perpendicular to the glass bottom. The larger segment of agar was removed and transferred to a fresh glass bottomed dish to reduce the amount of residual media on the glass surface. We prepared $A_{600} = 1.0$ optical units of cell suspension and inoculated the the agar and glass interface with three, evenly distributed, 1.2 μL drops of cell suspension and left to dry for 10 min. The dish was closed and incubated at 30 °C with the dish standing upright and the inoculated surface kept horizontal. After 7 h the dish was wrapped with Parafilm and placed into the microscope for imaging.

To record the development of live biofilm growth we used a Leica TCS SP8 DIM8 confocal microscope (Leica Microsystems, UK) fitted with HyD SMD detectors and equipped with solid-state lasers, 448 nm for CFP, 514 nm for YFP, and 552 nm for RFP.

A HC PL APO CS2 ×100 1.44NA oil immersion objective (Leica order number 11506325) was used with the digital zoom set to 1.5. We used line sequential bi-directional scanning with speed 200 Hz and a line average of 3 was used for RFP and 2 for YFP. The pinhole was set to 1.0 Airy units. Images were recorded as 12 bit images. The hardware autofocus was set on an empty region of the dish near to the positions imaged. The stage was contained in a temperature controlled case kept at 30 °C. The objective was cooled using the built in liquid cooling system to keep it at 28 °C.

**Fluorescent protein bleed through and background subtraction.** For all datasets we corrected for the mean background signal recorded by the microscopes. For the agarose pad time-lapse images this represents the background signal of the agar and media. For the low magnification biofilm cryosections it represents the background signal of the cryostat specimen matrix. For the high magnification cryosections it represents the cryostat specimen matrix and mounting medium.

For the low and high magnification biofilm cryosections we also calculated the contribution of the cell's natural autofluorescence (using the strain NCIB3610 with no fluorescent reporters) and bleed-through caused by activation of the RFP protein by the 512 nm laser (using the $P_{sigA}$-RFP reporter strain with no YFP reporter). The difference between NCIB3610 autofluorescence in the yellow channel and the mean yellow channel signal of an RFP only strain was used to quantify the bleed-through of the RFP protein into the YFP channel. For the high magnification images the bleed-through accounts for 6% of the YFP signal in the top of 20 μm of the central part of the WT biofilm at the 72 h time point. For the low magnification images the bleed-through accounts for 1.5% of the YFP signal in the top of 20 μm of the central part of the WT biofilm at the 72 h time point.

**Low magnification $\sigma^B$ gradients.** We extracted fluorescence gradients from confocal images of thin cryoslices of biofilm. The biofilm was identified in the red channel where cells were constitutively expressing $P_{sigA}$-RFP. The biofilm was segmented using custom Python software and the Scikit-Image package[77]. The biofilm is easily isolated with an Otzu threshold, however, this initial mask contained long strips of auto-fluorescent agar. To remove the agar we breakup the mask into sub-segments with similar values using a watershed segmentation with local maxima as seeds. We then erode the resulting over-segmentation with a circular selection element with a radius large enough to remove the sub-segments of auto-fluorescent agar (in our case 25 pixels). Any segments that are totally removed by the erosion are assumed to be agar and not restored, any other segments are restored to their pre-erosion shape. All contiguous segments are re-joined into a single biofilm mask. Any segments with area <10% of the image are removed and assumed to be artifacts of cryosectioning.

**Single cell and spore segmentation.** To segment both individual cells and individual spores from high magnification confocal images we used custom Python software and the Scikit-Image package[77]. The input images were either constitutive $P_{sigA}$-RFP for cells or a late stage sporulation $P_{sspB}$-YFP reporter.

**Spore density.** To quantify spore density we used tile-scan confocal images of the central section of the biofilm with single cell and spore resolution. For each image we calculated the number of spores and the number of cells in each depth bin $d \pm 1$ μm, for each $d$ sampled every 0.5 from 2 μm from the top of the biofilm to 140 μm. Then we plot either the ratio of spores to the combined number of cells and spores in each bin (see Fig. 7), or we scale the spore counts in each bin by the number of image pixels in that bin (see Supplementary Fig. 10B).

**Model details.** To explore the interactions of the $\sigma^B$ and the sporulation pathways, we used a qualitative model that produces stochastic pulses. Note that we do not attempt to model the biochemical processes that lead to the generation of $\sigma^B$ or sporulation pulses[55–58]. Instead, we use a variation of the well studied[68] two-step transcription–translation gene expression model used to study stochastic cell decision making and mutual inhibition. System A is reaction (1) to reaction (4), and system B is reaction (5) to reaction (8). Interaction between the two systems occurs in reaction (9) with B blocking the pulse initiation of A.

The full reaction network is as follows:

$$G_A \xrightarrow{\zeta s_A} G_A + R_A \tag{1}$$

$$R_A \xrightarrow{\beta} R_A + A \tag{2}$$

$$R_A \xrightarrow{\delta} \emptyset \tag{3}$$

$$A \xrightarrow{\epsilon} \emptyset \tag{4}$$

$$G_B \xrightarrow{\zeta s_B} G_B + R_B \tag{5}$$

$$R_B \xrightarrow{\beta} R_B + B \tag{6}$$

$$R_B \xrightarrow{\delta} \emptyset \tag{7}$$

$$B \xrightarrow{\epsilon} \emptyset \tag{8}$$

$$B + G_A \underset{u_B}{\overset{bB}{\rightleftharpoons}} BG_A \tag{9}$$

where $G_X$, $R_X$, $X$ represent the "gene", "mRNA" and "protein" of $X \in \{A, B\}$.

The time scale of the model is in seconds. We modified the equations and parameters from those used previously to examine bistable switches[68]. The parameters $\beta$, $\delta$, $\epsilon$, $b_B$, and $u_B$ are unchanged from ref. [68] and are estimations of the rate of the biological processes they represent. To avoid bistability, we lower the transcription rates $\zeta s_A$ and $\zeta s_B$. The parameters $s_A$, $s_B$ and $\zeta$ are chosen so that $A$ with reaction (1) at rate $\zeta s_A$ and reaction (5) at rate $2s_B\zeta$ for high $\zeta$ (high stress) are below the point where system $A$ dominates (simulations always produce a spore) or system $B$ dominates (simulations never produce a spore). At high and low $\zeta$, both species are still exhibiting pulsatile expression. The full list of parameters can be seen in Table 1.

Simulations used the Doob–Gillespie stochastic simulation algorithm[78]. Simulations were run from initial conditions where all species were absent except $G_A = G_B = 1$. Each simulation was allowed to run for 6.5 simulation minutes were discarded to ensure the system was out of the transient phase. To simulate the effect of the stress gradient we set $\zeta = 10^g$, where $g$ is one of 20 evenly sampled

**Table 1 Model parameter list.**

| Name | Value or range | Description |
|---|---|---|
| $\beta$ | $0.05 \, \text{s}^{-1}$ | Translation rate |
| $s_A$ | $0.7 \, \text{s}^{-1}$ | Transcription rate for $R_A$ |
| $s_B$ | $0.25 \, \text{s}^{-1}$ or $0.5 \, \text{s}^{-1}$ | Transcription rate for $R_B$ |
| $\zeta$ | $5 \times 10^{-5}$ to $5 \times 10^{-3}$ | Stress or gradient position |
| $\delta$ | $0.005 \, \text{s}^{-1} \, R_X^{-1}$ | mRNA degradation rate |
| $\epsilon$ | $5 \times 10^{-3} \, \text{s}^{-1} X^{-1}$ | Protein degradation rate |
| $b_B$ | $1.0 \, \text{s}^{-1} X^{-1}$ | $B$ binding to $G_A$ rate |
| $u_B$ | $0.1 \, \text{s}^{-1}$ | $B$ unbinding rate |

points from the interval $[\log_{10} 5 \times 10^{-5}, \log_{10} 5 \times 10^{-3}]$ representing the bottom $(5 \times 10^{-5})$ and top $(5 \times 10^{-3})$ of the biofilm. For each $\zeta$ we ran 10,000 simulations. For each simulation we recorded both: the final $B$ value and the longest duration species $A$ spent above a threshold $\theta = 1$. The threshold $\theta$ is the 20th percentile value of steady-state distribution of $A$ when $G_B = 0$ (estimated using the $\Gamma$ distribution approximation of this system[79]) and scales with the rise in pulse initiation rate which allows us to detect pulses at low initiation rates and in high initiation rates. For all the parameters used in this paper $\theta = 1$. A simulation is judged to have produced a spore if the longest duration $A$ was above its threshold is >30 simulated minutes.

## Data availability
Confocal images are available on request to the authors. The quantification of the confocal images are available at https://doi.org/10.5281/zenodo.3544513.

## Code availability
The analysis code and simulation code developed for this project is available at https://gitlab.com/slcu/teamJL/nadezhdin_murphy_et_al_2019.

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

## Acknowledgements

Dr. Avigdor Eldar (Tel-Aviv University) for the *B. subtilis* P$_{spoIID}$-CFP reporter strain. Dr. Nicola Stanley-Wall (The University of Dundee) for providing SPP1 phage culture and help and advise with the transduction protocol. Cambridge Advanced Imaging Centre for the assistance with samples preparation. Thanks to Dr. Raymond Wightman for extensive advice, troubleshooting, and help to capture timelapse confocal images of living biofilms.

## Author contributions

E.N., N.M., J.C.W.L. conceived and designed the study. E.N., N.M., J.C.W.L., and A.P. analysed and interpreted the data and wrote the article. E.N. constructed strains and performed the experiments. E.N. and N.M. developed the live biofilm imaging assay. N. M. implemented the model with input from A.P. and N.D. All authors provided input into the manuscript.

## Competing interests

The authors declare no competing interests.
