## [Peer Review File · Nature Communications]

Reviewers' comments:

Reviewer #1 (Remarks to the Author):

This ms examines the role of gene expression noise in facilitating the emergence of different cell types during the formation of a bacterial biofilm. The ms is well written, the experiments conducted carefully and well-controlled and innovative techniques are presented. I do, however, have a question regarding the interpretation presented, specifically, whether the observations really establish a causal relationship (as the title suggests) between noise and phenotypic differentiation. In addition, I have concerns about the reporter used.

1. The authors claim that noise in sigB pulsing "enables" the "top of the biofilm to consist of both spores and sigB active cells" (line 25-6). The experiment in Fig. 7 where they show that increasing the copy # of the phosphatase, a situation known to increase pulsing, decreases the frequency of spores in the top of the biofilm is consistent. However, a more definitive experiment would be to show that decreasing of noise in sigB pulsing has a positive impact on sporulation. Since the authors do not understand the source of noise (e.g., is it intrinsic or extrinsic?) such a manipulation is not possible. A hint that this experiment would not yield the expected result comes from the statement that "deleting sigB resulted in a qualitatively similar sporulation gradient as compared to WT" (line 209-11). Obviously, a deletion of the gene is not equivalent to reducing its "noise" but the authors should directly demonstrate that changes in the noise are necessary for the spatial phenotypic partitioning they observe.
2. The statement is made that "increasing phosphatase levels was previously shown to increase sB pulsing activity in microcolonies" (line 232) but what is the evidence that this is also so in the rather different situation of a biofilm?
3. The reporter used in this study (PsigB-yfp) does reflect the levels of sigB activation as it is dependent on SigB, but a more appropriate promoter would be of a downstream gene that is under control of SigB. The key question is: Are the pulses of sigB sufficient to activate downstream genes such that they have physiological effects? The aim of this paper (as opposed to previous ones about pulsing) as I understand is not to show simply that there is pulsing, but that in fact this pulsing has physiological consequences. Notably, the reporter chosen to assay sporulation (PsspB-yfp) is that of a downstream target of the sporulation sigma factor SigG, not the PsigG promoter itself.
4. An additional reason to avoid the sigB promoter itself as a reporter for sigB activity is, as the authors acknowledge (lines 85-8), "A key aspect of the generation of these pulses is that sB activates its own operon, consisting of sB, RsbV and RsbW. This results in both a positive feedback loop through the activation of sB, which amplifies fluctuations, and a negative feedback loop through the activation of RsbW, which terminates the pulse." Thus, measurement of the PsigB-YFP is reflective of a complicated set of parameters and confounds a simple interpretation of changes in the reporters.
5. The authors mentions that they avoided sigB activation by the laser by using a delta rbsRU mutation but why not a delta ytvA mutation?
6. The authors should look into using a 3610 derivative that is competent (J Bacteriol. 2013 Sep;195(18):4085-93).

Reviewer #2 (Remarks to the Author):

The manuscript by Nadezhdin, et al. examines the role of noise in *B. subtilis* biofilms. The work determines that noise in sigB expression can be heterogenous and lead to two opposing cell states: sporulating or entering a high sigB activated state, and that these cells exist together. Here the authors showed (1) sigB expression was heterogenous and pulses in single cells, (2) noise in expression is only dependent on energy stress, and (3) pulse activation through σ B allows for distinct sporulation and high sigB state, and these results were recapitulated using a simple

computational model. In addition, the manuscript introduces a new single-cell time-lapse protocol for studying single cells inside biofilms on solid surfaces. Overall, the manuscript is intriguing with proper controls and interesting results. It nicely complements recently published papers in the field of noise, dynamics, and single-cell effects in bacterial biofilms. I have a number of minor remarks to improve the quality and clarity of the work.

Minor concerns

1. Although energy and environmental stress pathways are of critical importance here, none of the experiments actually expose the cells to stress. It would be helpful to discuss how the introduction of either type of stress would manifest itself in the biofilm context. In addition, although the energy and environmental stress ideas are described in detail in previous work by the corresponding author, it would be useful in this manuscript to briefly describe what causes them, for instance what are conditions that would produce this stress that are relevant to biofilms.
2. The biofilm visualization assay described in Fig. 4A is nice, but it is not entirely clear to me that what is being imaged is the top of the biofilm and not its edge. This distinction is important in the context of other recent studies on bacterial biofilms that show interesting dynamics at the periphery of the biofilm. The authors should discuss any implications this may have.
3. Distinct from the previous point, it would be helpful to have a schematic showing what the top, center, etc. of the biofilm are (e.g. line 138 refers to the center of the biofilm, but which direction is it cut through).
4. Fig. 5: Are these data actually anticorrelated or are they just uncorrelated? Also, the legend lists >300k cells. Was it really possible to extract data from that many cells from two images?
5. Fig S6., is never mentioned or discussed in text. The ordering of Fig. S6 and Fig. S7 and the discussion of the 2xrsbQP strain in Fig. 7 needs attention.
6. Line 129: "delRU"
7. Fig. 3 and Fig. S3 and possibly elsewhere: it would be helpful to include the Δ in front of the gene name to avoid confusion and for consistency across the manuscript.
8. Line 216-217: What does "competing pulse frequencies" mean? Were different frequencies of pulsing tested?
9. Fig. 6B: The molecule numbers are low, typically <15 per cell. Is this realistic for the sigB system?
10. Last paragraph of Results, please clarify: the energy stress pathway has lower sigB activation and therefore higher sporulation? But why would there be more spores for the opposite strain? Line 231-233, makes it seem like the 2xrsbQP has higher sigB activity.
11. Gene names should be italicized throughout the text and figures.
12. Line 335: "LMT"
13. Methods, please check whether magnification of objective is included for single cell measurements. I only saw 20x, but may have missed it.
14. To simplify reading, I recommend consistent placement of the rsbQP and rsbRU data relative to each other. For instance, Fig. 2 uses rsbRU first while Fig. 3 presents rsbQP first. There are other places in the text and figures where this comes up as well, and it would be useful to just be consistent about the ordering.

Reviewer #3 (Remarks to the Author):

Nadezhdin et al. investigate the spatial and temporal activity of sigB transcription – the sigma factor responsible for the general stress response - in a biofilm and its relationship to the activation of another stress response, sporulation. These stress responses are sensitive to "energy stress" and "starvation stress", respectively. In a biofilm that is nourished from the bottom, gradients in both kinds of stresses will emerge and thus one expects that both the stress responses get induced at the top of the biofilm.

The experiments by Nadezhdin et al. confirm this expectation – both sigB promoter activity and sporulation increase towards the top of the biofilm (Fig.1- Fig.5). Moreover, the sigB promoter is known to pulse under energy stress in planar micro-colonies. Using a clever imaging strategy the authors succeed to show pulsatile activity in a 3D-setting of a biofilm. This is a nice technical achievement.

Earlier reports suggest that sigB inhibits sporulation. By increasing the gene dosage of the rbsQP operon, P_{sigB} activity increases and sporulation is globally reduced. Moreover, the maximum of sporulation within the biofilm appears to be shifted away from the top towards the center (Fig.7). This is a nice demonstration of how perturbations to stress signaling results in a globally altered biofilm structure.

From the abstract, I was hoping to learn more on “HOW the two regulators avoid competition in gene expression.” A mathematical model of competition by pulsing regulators (Fig. 6) is a first step. Nevertheless, I would encourage the authors to provide more MECHANISTIC INSIGHT into the proposed coupling between sigB-pulsing and (pulsatile) sporulation (initiation) from an experimental point of view. This could leverage the paper from “we propose” to “we show” that “stochastic pulsing of sigB allows cell to either activate sigB or sporulation”.

Below please find a list of additional comments/questions.

Introduction: Making references to spatial gradients in environmental conditions in biofilms - and as a result in stress signaling – already in the introduction, rather than the discussion - could help to place this study in proper context from the beginning. This could help to avoid confusion with cell-cell signaling during multi-cellular development of higher organisms, which is not addressed in this study.

Results:

Fig. 1 (pulsing of P_{sigB}):

- To compare P_{sigB} to P_{sigA} activities, both promoters should be ideally fused to the same fluorophore to exclude effects from the different nature of the fluorophores. mCherry and YFP- show substantial differences in stability under sporulation conditions. This could mask pulsatile dynamics and bias the distribution of fluorescence.
- The movie shows events from 6 hours and later, the tracking is performed for 0 – 4 hours. Both timescales should match each other.
- Why were frames of less than eight cells excluded from quantification of the histograms?
- Italicize genes in the cartoon with small letters.

Fig. 2 (spatial gradients):

- Why was sigW chosen as a “control”?
- From the biology of sigW activation, it was not obvious to me of whether there should be any sigW activity under biofilm conditions. Is there really more than autofluorescence? How much more? Is that sufficient activity to claim the absence of a gradient?
- Please confirm the γ -name of the gene and/or check your sigW construct. On Subtiwiki (and on BsubCyc) ybdS is linked to a beta-lactamase ybxB not to sigW (which is ybbL).

Fig. 4 (Pulsing in biofilms)

- 4D: Is normalization to the P_{sigA}-mCherry signal required to see the pulses? Why was this normalization applied?
- P_{sigA}-mCherry control for showing non-pulsing should be repeated with P_{sigA}-YFP.

Sporulation (and relationship to sigB), Fig.5 (and also 7)

- I am very confused and concerned by the SIGB KNOCKOUT DATA. If deleting sigB does not change sporulation, why do the authors believe that the pulsatile activity of sigB transcription is relevant at all for shaping sporulation in a wt biofilm?

Other questions to the coupling:

- What happens to sigB activity in sporulation mutant(s)?
- Why was PspoIID chosen as a sporulation marker? Would a promoter that is under control of Spo0A (which presumably - based on data in micro-colonies - should also show pulsatile activity) make the better choice for investigating the relationship between sigB and sporulation pathways in the context of the model in Fig. 6?

Model (Fig.6):

- Fig. 6B: Potential for confusion: The grey dashed line denotes the sporulation threshold. Why are the trajectories continued if this threshold is reached?

Spatial reorganization of stress responses in a mutant biofilm

(Fig.7):

- Compared to Fig.2 the spatial area that is shown is quite limited. Why?
- I have difficulties in interpreting Fig. 7D/C.
 - oWhy are there fluorescence dots at the bottom (?) of the "wt" biofilm (7C). What do they label? Spores?
 - oIn some part of the mutant biofilm, spores still seem to be present primarily at the top, in others at the bottom, rather than the center. This is not reflected in 7B, is it?
- How stable is the shift in position of the sporulation peak within the biofilm?
- What are other possibilities to modulate with the coupling between both pathways and how do they affect the results?

Materials and Methods

- It would be helpful to have some details on the fluorescence quantification moved to the main text.
- It would be helpful to include some details on the model in the main text.

Point by Point reply to Reviewers' comments

We thank the reviewers for their very helpful comments. We have carefully addressed them and believe the manuscript is much improved. In particular we now provide further analysis of the data showing support for the role of sigB pulsing in enabling phenotypic differentiation, as well as several control experiments for our reporter constructs. We have also provided new data showing that sigB gradients form on biofilms growing on plant roots, revealing that our results are relevant to a range of environmental conditions.

Please find below our point by point response to the specific comments.

Reviewer #1 (Remarks to the Author)

This ms examines the role of gene expression noise in facilitating the emergence of different cell types during the formation of a bacterial biofilm. The ms is well written, the experiments conducted carefully and well-controlled and innovative techniques are presented. I do, however, have a question regarding the interpretation presented, specifically, whether the observations really establish a causal relationship (as the title suggests) between noise and phenotypic differentiation. In addition, I have concerns about the reporter used.

We thank the reviewer for their positive assessment of our work. We have addressed their concerns about the role of noise and phenotypic differentiation, as well as the reporters used, below

1. The authors claim that noise in sigB pulsing “enables” the “top of the biofilm to consist of both spores and sigB active cells” (line 25-6). The experiment in Fig. 7 where they show that increasing the copy # of the phosphatase, a situation known to increase pulsing, decreases the frequency of spores in the top of the biofilm is consistent. However, a more definitive experiment would be to show that decreasing of noise in sigB pulsing has a positive impact on sporulation. Since the authors do not understand the source of noise (e.g., is it intrinsic or extrinsic?) such a manipulation is not possible. A hint that this experiment would not yield the expected result comes from the statement that “deleting sigB resulted in a qualitatively similar sporulation gradient as compared to WT” (line 209-11). Obviously, a deletion of the gene is not equivalent to reducing its “noise” but the authors should directly demonstrate that changes in the noise are necessary for the spatial phenotypic partitioning they observe.

We agree with the reviewer that manipulation of the noise source is not possible, and as such it is difficult to show that noise is driving the spatial phenotypic partitioning. The main claim of the paper, however, is that sigB pulsing (which is itself noisy) drives the spatial pattern, by allowing sporulation and sigB activation to occur in the same layer of the biofilm. Pulsed activation of sigB means that both sigB expressing cells and sporulating cells can co-exist at the top of the biofilm, even though the expression of each pathway is anti-correlated in individual cells due to sigB repressing sporulation. This is because only a fraction of cells experience the high sigB state at any one time. We have improved our analysis of the anti-correlation between sigB and sporulation Figure S9) that greatly strengthens this claim, and have improved our discussion of this point in the paper. This, in combination with our data showing that an increase in SigB pulsing activity (through the addition of the second

copy of the energy stress pathway RsbQP), blocks sporulation at the top of the biofilm, shows the role of noisy pulses of sigB in generating the patterns observed.

In the new version of the manuscript, we have rewritten the text in multiple places to clarify that we are discussing noisy pulses of gene expression, rather than solely noise in gene expression. We have also changed the title of our work to better reflect the main claim of our paper, to 'Stochastic pulsing of gene expression enables the generation of spatial patterns in *B. subtilis* Biofilms'.

Finally, we also agree with the reviewer that a deletion of the gene is not equivalent to reducing its noise. Furthermore, the observation that deleting the sigB gene does not significantly alter the sporulation gradient is consistent with our findings, since it indicates that the level of pulsing in the WT system is insufficient to substantially perturb the sporulation pattern, allowing both sporulation and sigB expression to co-exist. Thus, pulsed sigB transcription allows only a fraction of WT cells to transiently enter a high sigB state and block sporulation in those cells, without interfering with the overall sporulation pattern. It is only when this pulsing is substantially increased that we begin to see a detrimental effect on the overall pattern of sporulation. We have explained this point more clearly by inserting more explanation in the following text.

Due to the long-tailed distribution of σ B expression values (Fig. 3), only a fraction of cells exist in the high σ B and low PspolID-CFP expression state (Fig. 5) at any one time. This suggests that pulsed σ B expression allows a fraction of cells to enter the high σ B state without shutting off sporulation in all cells, which would occur if σ B was expressed in a non-pulsatile manner to a high level in all cells at the top of the biofilm. Deleting σ B resulted in a qualitatively similar sporulation gradient as compared to WT, confirming that the pulsing of σ B allows a proportion of cells to have levels of σ B that represses sporulation without interfering with the overall pattern of sporulation in the biofilm (Fig. S10).

We have also added more text to the following paragraph:

We examined the effect of increasing RsbQP copy number on the σ B gradient, and on the distribution of spore forming locations. As previously observed in liquid culture [29], we observed higher single cell σ B expression in the 2xrsbQP strain than WT, with the single cell distributions remaining heterogeneous, characteristic of pulsing. Pulsing could also still be observed in timelapse movies of the 2xrsbQP strain (Sup Movie S3). We then addressed the effects of the higher σ B activity during biofilm formation. The 2xrsbQP strain has a much stronger σ B gradient than WT (Fig. 7.B S11.A) and more extreme heterogeneity in σ B expression (Fig. S11.B). In 2xrsbQP the spore forming region is shifted away from the top of the biofilm (brown line, Fig. 7.A, D) compared to WT (Fig. 7.A, C), confirming that σ B is repressing sporulation in biofilms, and that pulsing of σ B in WT cells allows the coexistence of both cell states.

2. The statement is made that "increasing phosphatase levels was previously shown to increase sB pulsing activity in microcolonies" (line 232) but what is the evidence that this is also so in the rather different situation of a biofilm?

In order to examine the effects of raising phosphatase levels through the addition of the second copy of the RsbQP pathway, in the first submission we analysed snapshots of gene expression from the biofilms at multiple timepoints in both WT and 2xRsbQP strains. We found that activity of sigB expression was increased by 80%, although the expression remained heterogeneous. This qualitatively matches the effects previously observed in microcolonies. Although this data shows that the 2xQP strain is generating higher sigB activity, we now include (Sup Movie S3) a supplementary movie showing that sigB remains pulsatile in the 2xQP strain, thereby

showing that the higher level of sigB expression we observe from our snapshots is due to the increased sigB activity. We have also improved our discussion of this point in the text.

As previously observed in liquid culture [29], we observed higher single cell σ_B expression in the 2xrsbQP strain than WT, with the single cell distributions remaining heterogeneous, characteristic of pulsing. Pulsing could also still be observed in timelapse movies of the 2xrsbQP strain (Sup Movie S3). We then addressed the effects of the higher σ_B activity during biofilm formation.

3. The reporter used in this study (PsigB-yfp) does reflect the levels of sigB activation as it is dependent on SigB, but a more appropriate promoter would be of a downstream gene that is under control of SigB. The key question is: Are the pulses of sigB sufficient to activate downstream genes such that they have physiological effects? The aim of this paper (as opposed to previous ones about pulsing) as I understand is not to show simply that there is pulsing, but that in fact this pulsing has physiological consequences. Notably, the reporter chosen to assay sporulation (PsspB-yfp) is that of a downstream target of the sporulation sigma factor SigG, not the PsigG promoter itself.

We apologise for the confusion. The promoter used in this study contains the upstream 200 bp before the start of the first gene (RsbV) in the sigB operon. This contains the sigB binding site from the sigB operon and no other sigma factor binding site, and does represent the activity of sigB, rather than the expression level. The expression level of sigB is controlled by both a sigA binding site (in front of RsbU in a previous operon) and the sigB binding site in front of RsbV. In previous work we demonstrated that this PsigB-YFP construct displays the same activation dynamics as multiple other sigB target genes in cells grown in liquid culture [1].

To strengthen this point, we have constructed new reporters of downstream genes activated by sigB in a strain of B. subtilis capable of generating biofilms. These strains, PcsbB-YFP and PyfIA-YFP, have heterogeneous expression, as well as a gradient in activation, in biofilm snapshots (Fig. S8). This further validates our choice of reporter for sigB activity in the context of the biofilm.

1. Locke, J. C. W.; Young, J. W.; Fontes, M.; Jiménez, M. J. H. & Elowitz, M. B. Stochastic Pulse Regulation in Bacterial Stress Response. *Science*, 2011 , 334 , 366-369

4. An additional reason to avoid the sigB promoter itself as a reporter for sigB activity is, as the authors acknowledge (lines 85-8), "A key aspect of the generation of these pulses is that sB activates its own operon, consisting of sB , RsbV and RsbW. This results in both a positive feedback loop through the activation of sB , which amplifies fluctuations, and a negative feedback loop through the activation of RsbW, which terminates the pulse." Thus, measurement of the PsigB-YFP is reflective of a complicated set of parameters and confounds a simple interpretation of changes in the reporters.

As above, the promoter for our reporter construct, PsigB-YFP contains a sigB binding site driving expression, so the expression of our reporter construct is modulated by active sigB. We have also validated our reporter with two additional reporter constructs, PcsbB-YFP and PyfIA-YFP.

5. The authors mentions that they avoided sigB activation by the laser by using a delta rbsRU mutation but why not a delta ytvA mutation?

A Delta ytvA mutation, as the reviewer correctly points out, would have avoided sigB activation due to blue light. The deltaRU mutation has the additional advantage that

any additional spurious environmental stress sigB activation due to our microscopy conditions would not affect our movie analysis.

6. The authors should look into using a 3610 derivative that is competent (J Bacteriol. 2013 Sep;195(18):4085-93).

Thank you for this suggestion, we have used this strain in constructing the additional new strains used in this resubmission.

Reviewer #2 (Remarks to the Author):

Reviewer 2

The manuscript by Nadezhdin, et al. examines the role of noise in *B. subtilis* biofilms. The work determines that noise in sigB expression can be heterogenous and lead to two opposing cell states: sporulating or entering a high sigB activated state, and that these cells exist together. Here the authors showed (1) sigB expression was heterogenous and pulses in single cells, (2) noise in expression is only dependent on energy stress, and 3) pulse activation through B allows for distinct sporulation and high sigB state, and these results were recapitulated using a simple computational model. In addition, the manuscript introduces a new single-cell time-lapse protocol for studying single cells inside biofilms on solid surfaces. Overall, the manuscript is intriguing with proper controls and interesting results. It nicely complements recently published papers in the field of noise, dynamics, and single-cell effects in bacterial biofilms. I have a number of minor remarks to improve the quality and clarity of the work.

We thank the reviewer for their positive assessment. We have addressed their minor concerns below

Minor concerns

1. Although energy and environmental stress pathways are of critical importance here, none of the experiments actually expose the cells to stress. It would be helpful to discuss how the introduction of either type of stress would manifest itself in the biofilm context. In addition, although the energy and environmental stress ideas are described in detail in previous work by the corresponding author, it would be useful in this manuscript to briefly describe what causes them, for instance what are conditions that would produce this stress that are relevant to biofilms.

We agree that this is an interesting point that should have more discussion in the manuscript. The energy stress pathway is activated in ATP limiting conditions. The biofilm media is a minimal media with only glycerol as a carbon source. As cells grow on top of the agarose pad containing MSgg medium, it is quite likely that cells will experience energy stress during biofilm growth. We now discuss how biofilm growth could expose the cells to energy stress at the top of the biofilm, and how this might explain the gradient in sigB expression, in more detail in the discussion:

We observe a gradient of σ_B activation, dependent on the energy stress pathway, at the top of the biofilm that could correspond to on a gradient of nutrients that depletes near the top of the biofilm.

Multiple environmental stresses (heat, ethanol, NaCl), known to activate SigB in liquid culture, could also activate sigB in cells during biofilm formation through external application. We have added the following text to the introduction that provides detail concerning the possible sources of energy and environmental stress.

Energy stresses include ATP limitation through the addition of inhibitors (e.g. CCCP, MPA), entry into stationary phase, or carbon limiting media. Environmental stresses

shown to activate σ_B include ethanol, NaCl and heat. These responses have been characterised in planktonic growth, and it is unclear what the activation dynamics are of σ_B during biofilm formation. Biofilms can consist of spatially localised stress patterns [37], as well as gradients of nutrients away from the nutrient source [38], and thus to understand σ_B activation during biofilm formation it is critical to examine gene expression in individual cells.

2. The biofilm visualization assay described in Fig. 4A is nice, but it is not entirely clear to me that what is being imaged is the top of the biofilm and not its edge. This distinction is important in the context of other recent studies on bacterial biofilms that show interesting dynamics at the periphery of the biofilm. The authors should discuss any implications this may have.

In the works the reviewer is referring to, and in the context of this work, the edge of the biofilm is a region with access to fresh media and fast cell growth at the edge of the colony. We have selected snapshot slices from the centre of the colony in Figure 2-3 in the paper. In Figure 4A movie conditions, we observe wrinkle formation consistent with the 'centre' morphology phenotype, and not the smooth edge morphology phenotype. Our movie data also matches well to the snapshot data we report from the centre of the colony in figures 2-3. Our imaging conditions are quite distinct from those used to study edge phenotype cells described in previous papers. We now include a clarification of this in the main text:

(We took our samples from the centre of the biofilm colony, which has a wrinkle morphology distinct from the smoother edge of the biofilm (Fig. S1).

and to make this more clear to readers we have added annotations to a supplemental Fig. S1.

3. Distinct from the previous point, it would be helpful to have a schematic showing what the top, center, etc. of the biofilm are (e.g. line 138 refers to the center of the biofilm, but which direction is it cut through).

We have added annotations to a supplemental figure (Fig. S1) showing the center and edge morphology of the biofilm. We also label what we consider the top and bottom. We hope this clears up any confusion.

4. Fig. 5: Are these data actually anticorrelated or are they just uncorrelated? Also, the legend lists >300k cells. Was it really possible to extract data from that many cells from two images?

We now provide further analysis showing the level of anticorrelation in figure 5. The images for the analysis were stitched tile-scans of large (~3mm) colony regions, whilst only a fraction of this image is depicted in Figure 5. We now clarify this in the figure legend with the text:

The data represent two tilescan images, each covering approximately 3mm of biofilm, from two different experiments and 372689 cells.

5. Fig S6., is never mentioned or discussed in text. The ordering of Fig. S6 and Fig. S7 and the discussion of the 2xrsbQP strain in Fig. 7 needs attention.

Thank you for noticing this, we have now referenced the figures in the text in the correct order.

6. Line 129: "delRU"

Fixed, thank you.

7. Fig. 3 and Fig. S3 and possibly elsewhere: it would be helpful to include the Δ in front of the gene name to avoid confusion and for consistency across the manuscript.

We have now done this consistently, thank you!

8. Line 216-217: What does “competing pulse frequencies” mean? Were different frequencies of pulsing tested?

We agree that this section was not clear. We have clarified this statement in the text, which now reads:

We tested how the pulsed activation of two competing pathways affected cell fate in our model

We do modulate sigB pulse frequency later in this section, but here we were attempting to describe the two competing pulsing systems.

9. Fig. 6B: The molecule numbers are low, typically <15 per cell. Is this realistic for the sigB system?

The model is an extremely simple model that represents two interacting pulsing systems, rather than a detailed model of the sigB network. The absolute molecule numbers are low in order to simulate noisy transcription. The numbers of molecules of sigB before and after stress are currently not known, although it will be important in future work to measure these accurately.

10. Last paragraph of Results, please clarify: the energy stress pathway has lower sigB activation and therefore higher sporulation? But why would there be more spores for the opposite strain? Line 231-233, makes it seem like the 2xrsbQP has higher sigB activity.

We apologise for the confusion caused by this section. The reviewer is correct that the 2xrsbQP line has higher sigB activity, which we find causes lower sporulation. We have added additional details to this section to avoid confusion for the reader:

As previously observed in liquid culture [29], we observed higher single cell σ_B expression in the 2xrsbQP strain than WT, with the single cell distributions remaining heterogeneous, characteristic of pulsing. Pulsing could also still be observed in timelapse movies of the 2xrsbQP strain (Sup Movie S3). We then addressed the effects of the higher σ_B activity during biofilm formation.

11. Gene names should be italicized throughout the text and figures.

We agree, and have fixed this throughout the text.

12. Line 335: “LMT”

We have expanded the acronym to Low Melting Temperature in the revised text.

13. Methods, please check whether magnification of objective is included for single cell measurements. I only saw 20x, but may have missed it.

The objective information was missing for the single cell pad movies (Figure 1). We now ensure the objective and magnification is defined in the materials and methods section.

14. To simplify reading, I recommend consistent placement of the rsbQP and rsbRU data relative to each other. For instance, Fig. 2 uses rsbRU first while Fig. 3 presents rsbQP first.

There are other places in the text and figures where this comes up as well, and it would be useful to just be consistent about the ordering.

We agree, and have standardized the placement of data in both the main and supplemental figures.

Reviewer #3 (Remarks to the Author):

Nadezhdin et al. investigate the spatial and temporal activity of sigB transcription – the sigma factor responsible for the general stress response - in a biofilm and its relationship to the activation of another stress response, sporulation. These stress responses are sensitive to “energy stress” and “starvation stress”, respectively. In a biofilm that is nourished from the bottom, gradients in both kinds of stresses will emerge and thus one expects that both the stress responses get induced at the top of the biofilm.

The experiments by Nadezhdin et al. confirm this expectation – both sigB promoter activity and sporulation increase towards the top of the biofilm (Fig.1- Fig.5). Moreover, the sigB promoter is known to pulse under energy stress in planar micro-colonies. Using a clever imaging strategy the authors succeed to show pulsatile activity in a 3D-setting of a biofilm. This is a nice technical achievement.

Earlier reports suggest that sigB inhibits sporulation. By increasing the gene dosage of the rbsQP operon, P_{sigB} activity increases and sporulation is globally reduced. Moreover, the maximum of sporulation within the biofilm appears to be shifted away from the top towards the center (Fig.7). This is a nice demonstration of how perturbations to stress signaling results in a globally altered biofilm structure.

Thank you for your positive assessment of our work. We have addressed your major and minor concerns below

From the abstract, I was hoping to learn more on “HOW the two regulators avoid competition in gene expression.” A mathematical model of competition by pulsing regulators (Fig. 6) is a first step. Nevertheless, I would encourage the authors to provide more MECHANISTIC INSIGHT into the proposed coupling between sigB-pulsing and (pulsatile) sporulation (initiation) from an experimental point of view. This could leverage the paper from “we propose” to “we show” that “stochastic pulsing of sigB allows cell to either activate sigB or sporulation”.

We agree with the reviewer that further insight into the mechanisms by which sigB represses sporulation would be valuable. These mechanisms have previously been characterised by others, and shown to act through the upregulation of spo0E, a negative regulator of the master regulator of sporulation [1,2,3]. We now discuss this mechanism in more detail in the text. Building on this known mechanism, our work focuses on proposing a role for noisy pulsing in pattern formation. To address the reviewer comments and show that stochastic pulsing of sigB allows cells to either activate sigB or sporulation, we have provided additional controls and experiments recommended by the reviewers, including characterising more carefully the anticorrelation between sigB and sporulation in individual WT cells, examining additional sigB reporter strains, verifying that heterogeneous sigB activity occurs in a gradient in a more ‘natural’ growth setting of biofilm growth on plant roots, and verifying that a sporulation deletion does not modulate sigB expression, as outlined below.

1. Reder, A.; Albrecht, D.; Gerth, U. & Hecker, M. Cross-talk between the general stress response and sporulation initiation in *Bacillus subtilis* - the B promoter of *spo0E* represents an AND-gate. *Environ. Microbiol.* 2012 , 14 , 2741-2756
2. Reder, A.; Gerth, U. & Hecker, M. Integration of B Activity into the Decision-Making Process of Sporulation Initiation in *Bacillus subtilis*. *J. Bacteriol.*, 2012 , 194 , 1065-1074
3. Rothstein, D. M.; Lazinski, D.; Osburne, M. S. & Sonenshein, A. L. A mutation in the *Bacillus subtilis* *rsbU* gene that limits RNA synthesis during sporulation. *J. Bacteriol.*, 2017 , 199 , e19131

Below please find a list of additional comments/questions.

Introduction: Making references to spatial gradients in environmental conditions in biofilms - and as a result in stress signaling – already in the introduction, rather than the discussion - could help to place this study in proper context from the beginning. This could help to avoid confusion with cell-cell signaling during multi-cellular development of higher organisms, which is not addressed in this study.

We agree, and have added the following text discussing this point in the introduction.

Energy stresses include ATP limitation through the addition of inhibitors (e.g. CCCP, MPA), entry into stationary phase, or carbon limiting media. Environmental stresses shown to activate σ_B include ethanol, NaCl and heat. These responses have been characterised in planktonic growth, and it is unclear what the activation dynamics are of σ_B during biofilm formation. Biofilms can consist of spatially localised stress patterns [37], as well as gradients of nutrients away from the nutrient source [38], and thus to understand σ_B activation during biofilm formation it is critical to examine gene expression in individual cells.

Results:

Fig. 1 (pulsing of P_{sigB}):

•To compare P_{sigB} to P_{sigA} activities, both promoters should be ideally fused to the same fluorophore to exclude effects from the different nature of the fluorophores. mCherry and YFP-show substantial differences in stability under sporulation conditions. This could mask pulsatile dynamics and bias the distribution of fluorescence.

We used P_{sigA}-RFP and P_{sigB}-YFP reporters to allow comparison of the two pathways in the same cell. We have previously observed that sigA regulated reporter driving YFP is homogenous under energy stress conditions that promote sigB pulsing in a sigB-YFP reporter [1]. However, we agree that it is important to verify that a P_{sigA}-YFP reporter is also homogeneous under our biofilm conditions. We generated a P_{sigA}-YFP reporter in our biofilm generating bacillus background, and confirmed that P_{sigA}-YFP is expressed constitutively throughout the biofilm, and has a homogeneous single cell distribution. We also observed strong correlation between a P_{sigA}-YFP and P_{sigA}-RFP dual reporter strain. We have added a new figure Fig S4 showing the results.

1. Park, Dies, Lin, Hormoz, Smith-Unna, Quinodoz, Hernández-Jiménez, Garcia-Ojalvo, Locke, Elowitz. Molecular Time Sharing through Dynamic Pulsing in Single Cells. *Cell Systems*, 2018 , 6 , 216-229.e15

The movie shows events from 6 hours and later, the tracking is performed for 0 – 4 hours. Both timescales should match each other.

We agree. The x-axis in the time series plots are each reset to zero at the start of a 4 hour window where we track the cells. The time stamps in subplot B were relative to the start of the experiment. We have now updated the film strip timestamps to match the plots. To clarify this we have added the following text to the caption of Fig 1.

The time in hours is relative to the start of analysis.

Why were frames of less than eight cells excluded from quantification of the histograms?

We apologize, this should have been explained in the text. We have added the following explanation to the section “Agarose pad time-lapse microscopy” in the Material and Methods.

When analysing single cell data we discard data from the first frames of the movie (before 8 cells) to allow cells to adjust to growth on the agarose pads and to avoid inclusion of cells with fluorescent protein levels that could reflect the previous planktonic growth or the transition to the agarose pad

Italicize genes in the cartoon with small letters.

Thank you, we have changed gene symbols to be italicized initial lowercase throughout the paper.

Fig. 2 (spatial gradients):

Why was sigW chosen as a “control”? From the biology of sigW activation, it was not obvious to me of whether there should be any sigW activity under biofilm conditions. Is there really more than autofluorescence? How much more? Is that sufficient activity to claim the absence of a gradient?

We chose sigW expression as a control because it is an alternative sigma factor, similar to sigB, and there are multiple published evidences of involvement of sigW in the process of biofilm formation. These include [1] also reviewed in [2]. We now clarify this in the paper with the following text and citations:

σ W (PydbS-YFP) which is active in biofilm formation [45, 46, 47].

[45] Cao, M.; Wang, T.; Ye, R. & Helmann, J. D. Antibiotics that inhibit cell wall biosynthesis induce expression of the Bacillus subtilis W and M regulons. Mol. Microbiol., 2002 , 45 , 1267-1276

[46] Butcher, B. G. & Helmann, J. D. Identification of Bacillus subtilis sigma-dependent genes that provide intrinsic resistance to antimicrobial compounds produced by Bacilli . Mol. Microbiol., 2006 , 60 , 765-782

[47] Helmann, J. D. Bacillus subtilis extracytoplasmic function (ECF) sigma factors and defense of the cell envelope. Curr. Opin. Microbiol., Elsevier Ltd, 2016 , 30 , 122-132

Our own data shows that the sigW reporter’s level of expression is much higher than autofluorescence, which we now demonstrate in an updated Fig. S3.

Please confirm the y-name of the gene and/or check your sigW construct. On Subtiwiki (and on BsubCyc) ybdS is linked to a beta-lactamase ybxI not to sigW (which is ybbL).

We note that the name of our reporter for sigW is ydbS, not ybdS, and is a well characterised reporter for sigW activity (Butcher, 2006) with the following Subtiwiki and BsubCyc links

<https://bsubcyc.org/gene?orgid=BSUB&id=BSU04590> and

<http://www.subtiwiki.uni-goettingen.de/v3/gene/view/E4139214AB491712309F84A0792BCD8939F634D7>

1. Huang, Fredrick, Helmann, "Promoter recognition by *Bacillus subtilis* sigmaW: autoregulation and partial overlap with the sigmaX regulon." *Journal of bacteriology*, 1998. PMID: 9683469.
2. Vlamakis, Chai, Beaugregard, Losick, Kolter. "Sticking together: building a biofilm the *Bacillus subtilis* way", *Nat. Rev. Microbiol.*, 2013 PMID: 23353768.
3. Butcher, Helmann. "Identification of *Bacillus subtilis* sigma-dependent genes that provide intrinsic resistance to antimicrobial compounds produced by *Bacilli*". *Molecular microbiology*. 2006. PMID:16629676.

Fig. 4 (Pulsing in biofilms)

4D: Is normalization to the P_sigA-mCherry signal required to see the pulses? Why was this normalization applied?

The normalization is not required. We apply this normalisation in figure 4 to show that YFP pulses occur even when normalized by the sigA-mCherry signal - ie the pulses are not due to fluctuations in constitutive gene expression. In the original submission we included Fig. S7 that did not have the normalization applied. We retain this figure.

P_sigA-mCherry control for showing non-pulsing should be repeated with P_sigA-YFP.

As discussed above, we have performed an additional experiment showing that P_sigA-YFP correlates with PsigA-RFP expression and does not display a gradient in gene expression is now in a Supplemental Figure S4. In addition, P_sigA-YFP distribution is homogeneous, matching P_SigA-RFP and indicating non-pulsing dynamics.

Sporulation (and relationship to sigB), Fig.5 (and also 7)

•I am very confused and concerned by the SIGB KNOCKOUT DATA. If deleting sigB does not change sporulation, why do the authors believe that the pulsatile activity of sigB transcription is relevant at all for shaping sporulation in a wt biofilm?

We apologise that this section was confusingly written. We have now re-written this, with additional analysis, to clarify our points. We now more accurately describe the anti-correlation between sigB and sporulation at the level of individual cells (Fig. S9). Our sigB knockout data shows that although some cells in a WT biofilm are in a high sigB state that represses sporulation, the pattern of sporulation is still able to peak at the top of the biofilm (as it does in the sigB knockout). In a 'deterministic scenario' - if all cells had the same level of high sigB in the top layer of the biofilm, this would block the sporulation pathway and interfere with the pattern. Thus pulsed sigB transcription allows a subset of cells to transiently enter high sigB state without interfering with the sporulation pattern. We do not propose that sigB is required for sporulation to peak at the top of the biofilm, as this appears intrinsic to the sporulation pathway. We have now clarified this in the text:

Due to the long-tailed distribution of σ_B expression values (Fig. 3), only a fraction of cells exist in the high σ_B and low PspolID-CFP expression state (Fig. 5) at any one time. This suggests that pulsed σ_B expression allows a fraction of cells to enter the high σ_B state without shutting off sporulation in all cells, which would occur if σ_B was expressed in a non-pulsatile manner to a high level in all cells at the top of the biofilm. Deleting σ_B resulted in a qualitatively similar sporulation

gradient as compared to WT, confirming that the pulsing of σ_B allows a proportion of cells to have levels of σ_B that represses sporulation without interfering with the overall pattern of sporulation in the biofilm (Fig. S10).

Other questions to the coupling:

What happens to sigB activity in sporulation mutant(s)?

Thank you for this interesting suggestion. We verified that sigB activity and the sigB gradient are not affected by a sporulation mutant (delSigF). We have added a new supplemental figure with this analysis (see Sup Fig. S8 A.d B.d C.d).

Why was PspolID chosen as a sporulation marker? Would a promoter that is under control of Spo0A (which presumably - based on data in micro-colonies - should also show pulsatile activity) make the better choice for investigating the relationship between sigB and sporulation pathways in the context of the model in Fig. 6?

We wanted to ascertain in this study the effect of sigB in successful sporulation. As not every activation of Spo0A controlled genes leads to sporulation, we chose spolID as it is under the control of sigE – an early mother cell sporulation sigma factor. Activation of sigE is a strong indication of commitment to sporulation [1]. We agree that in future work that it will be important to also examine early sporulation dynamics.

1. Narula, J.; Devi, S. N.; Fujita, M. & Igoshin, O. A. Ultrasensitivity of the Bacillus subtilis sporulation decision. PNAS, 2012 , 109 , E3513-E3522

Model (Fig.6):

Fig. 6B: Potential for confusion: The grey dashed line denotes the sporulation threshold. Why are the trajectories continued if this threshold is reached?

We apologise for the confusion. We clarify that Figure 6B aims to demonstrate the mutually exclusive time series produced by the model. In our simulation a cell must have A over the threshold for 30 simulated minutes before it is considered a spore. In this trace the cell does not spend more than 30 minutes over the threshold so it does not become a spore. We have added the following text to the figure caption to clarify.

The gray dashed line is the spore threshold (for these parameters, a cell must express A for more than 30 simulated minutes to become a spore. For details see Materials and methods).

Spatial reorganization of stress responses in a mutant biofilm (Fig. 7):

Compared to Fig.2 the spatial area that is shown is quite limited. Why?

This set of images had a number of technical artifacts from the cryoslicing. We chose a region with as few artifacts as possible so to not confuse the reader. However, the phenotype can actually be clearly seen with the full images so we now include whole images as a new supplemental Figure S12.

I have difficulties in interpreting Fig. 7D/C. Why are there fluorescence dots at the bottom (?) of the "wt" biofilm (7C). What do they label? Spores?

We do observe spores below the biofilm at the agar--colony interface. The area below the biofilm was not included in the analysis due to the difficulties in setting the agar--colony interface and the lack of cells in this region, but we agree this should have been explicitly commented on in the paper.

We have added text to the caption to address this:

“Note that in (C), between the biofilm and agar, is a layer of spores. This is typical of WT biofilms and is not reflected in the spore gradient (A) since there are no cells to compute a ratio. The full context for (C) and (D) is shown in Fig. S12. “

In some part of the mutant biofilm, spores still seem to be present primarily at the top, in others at the bottom, rather than the center. This is not reflected in 7B, is it?

Figure 7.B shows the P_{sigB}-YFP gradient so does not reflect what we see in 7.C or 7.D. To avoid confusion we have added a title to each subfigure indicating if it is depicting spores or sigmaB.

The 2xQP spore gradient in Figure 7A however does reflect what we see in 7.C or 7.D. It contains the data of 7 images of large tile scans each covering several millimeters of 3 different biofilms. Small regions where there are more spores at the top or bottom are included in this analysis. To allow readers to assess the robustness of the result more easily, we now include a complete tile scan as a supplemental Figure S12.

How stable is the shift in position of the sporulation peak within the biofilm?

The shift of the sporulation peak in the 2xQP strain is quite robust - as mentioned above we now include the full tilescan in a new supplemental Figure S12 to show this, as well as new analysis measuring the sporulation peak across all our experiments (Figure S10.C).

What are other possibilities to modulate with the coupling between both pathways and how do they affect the results?

We now include new results showing that the sigB pathway is expressed in a gradient in biofilms on plant roots (Fig. S5). As it has been shown (Beauregard 2013) that *B. subtilis* is often found in association with plant roots in soil. Fig S5 shows that sigB is expressed in gradients in a more ‘natural’ growth setting, as well as on agar plates. This raises the possibility of examining the coupling between sigB and sporulation in other growth contexts (such as roots), although we would consider this as future work and outside the scope of this paper.

- 1. Beauregard, Chai, Vlamakis, Losick, Kolter. *Bacillus subtilis* biofilm induction by plant polysaccharides. PNAS 1621-30, 110, 2013, doi: 10.1073/pnas.1218984110**

Materials and Methods

It would be helpful to have some details on the fluorescence quantification moved to the main text.

We agree, and have moved materials and methods related to the quantification from the supplementary methods into the methods in the main text.

It would be helpful to include some details on the model in the main text.

We agree, and now include more details about the model in the materials and methods in the main text rather than the supplemental text.

REVIEWERS' COMMENTS:

Reviewer #1 (Remarks to the Author):

The authors have appropriately answered my critique

Reviewer #3 (Remarks to the Author):

The authors have satisfactorily addressed my questions and concerns and the revised manuscript is more easy to follow. Especially the nature of the coupling has become more clear. I advocate for publication.

Two minor remarks:

1. Just to make sure that I got the final message, regarding the coupling:

Sporulation does not affect sigB but high levels of sigB inhibit sporulation. Hence the coupling is unidirectional.

I suppose that unidirectional coupling requires pulsing in order to generate coexisting states in a system? (This is in contrast to double negative couplings which could generate coexisting states by noise driven bifurcations based on a bistable system)?

Perhaps a sentence along those lines could be added to the discussion if the authors find it to be appropriate.

2. Description of the sporulation mutant: Consider to replace "sporulation pathway" by "sporulation" or alternatively, be more specific to explain where sporulation is blocked. Since the model considers pulses of A (spo0A) and B (sigB), I consider it to be an important detail that Spo0A-signaling is still intact in the sporulation mutant studied.

Response to Reviewers.

The authors have satisfactorily addressed my questions and concerns and the revised manuscript is more easy to follow. Especially the nature of the coupling has become more clear. I advocate for publication.

Thank you for your positive assessment of our revisions.

Two minor remarks:

1. Just to make sure that I got the final message, regarding the coupling:

Sporulation does not affect sigB but high levels of sigB inhibit sporulation. Hence the coupling is unidirectional.

I suppose that unidirectional coupling requires pulsing in order to generate coexisting states in a system? (This is in contrast to double negative couplings which could generate coexisting states by noise driven bifurcations based on a bistable system)?

Perhaps a sentence along those lines could be added to the discussion if the authors find it to be appropriate.

The reviewer is correct that we find that the coupling is unidirectional. We added the following sentence to the discussion:

'Going forward, it will be interesting to use such a model to test the differences between unidirectional coupling and a mutually repressive bistable switch circuit [1].'

[1] Stability and Multiattractor Dynamics of a Toggle Switch Based on a Two-Stage Model of Stochastic Gene Expression. Biophys. J., 2012, Michael Strasser and Fabian J. Theis and Carsten Marr <http://dx.doi.org/10.1016/j.bpj.2011.11.4000>

2. Description of the sporulation mutant: Consider to replace "sporulation pathway" by "sporulation" or alternatively, be more specific to explain where sporulation is blocked. Since the model considers pulses of A (spo0A) and B (sigB), I consider it to be an important detail that Spo0A-signaling is still intact in the sporulation mutant studied.

We have updated the text so that it now reads:

'We first confirmed that sporulation is not required for the gradient in σ_B expression (Supplementary Fig. 8.D) by deleting σ_F , which is required for spore formation.'